# Calibrating
# "Cheap Signals" in Peer Review without a Prior

**Yuxuan Lu**
Center on Frontiers of Computing Studies
School of Computer Science
Peking University
Beijing, China
yx_lu@pku.edu.cn

**Yuqing Kong**[*]
Center on Frontiers of Computing Studies
School of Computer Science
Peking University
Beijing, China
yuqing.kong@pku.edu.cn

## Abstract

Peer review lies at the core of the academic process, but even well-intentioned reviewers can still provide noisy ratings. While ranking papers by average ratings may reduce noise, varying noise levels and systematic biases stemming from "cheap" signals (e.g. author identity, proof length) can lead to unfairness. Detecting and correcting bias is challenging, as ratings are subjective and unverifiable. Unlike previous works relying on prior knowledge or historical data, we propose a one-shot noise calibration process without any prior information. We ask reviewers to predict others' scores and use these predictions for calibration. Assuming reviewers adjust their predictions according to the noise, we demonstrate that the calibrated score results in a more robust ranking compared to average ratings, even with varying noise levels and biases. In detail, we show that the error probability of the calibrated score approaches zero as the number of reviewers increases and is significantly lower compared to average ratings when the number of reviewers is small.

## 1 Introduction

Peer review is fundamental to the scientific process. However, as noted in Rennie [1]:

*But it is a human system. Everybody involved brings prejudices, misunderstandings, and gaps in knowledge, so no one should be surprised that peer review is often **biased** and inefficient ... even with **the best of intentions** ...*

Let us consider the following scenario.

**Example 1** (Peer review). *We have two papers and each paper is reviewed by 5 reviewers. We can only accept one paper. The first paper receives 4 "accept" and 1 "reject", the second paper receives 3 "accept" and 2 "reject".*

A typical choice is to accept the first paper since it receives more fraction of "accept". Ideally, we should accept the paper which receives more fraction of "accept" in expectation from reviewers who have full expertise and invest full efforts, called the clean setting. However, in practice, the ratings can be noisy due to the lack of expertise or effort. If the noise is consistent for the two papers, the better paper should still receive more fraction of "accept" in expectation. Nevertheless, the noise can be different for different papers.

**Example 2** (Hot vs. cold topic). *The topic of the first paper is more popular than the second paper. In this case, the first paper is easier to obtain reviewers with high expertise. Thus, in the noisy setting, the first paper has less noisy ratings than the second paper.*

---

[*]Corresponding author.

37th Conference on Neural Information Processing Systems (NeurIPS 2023).

It is possible that in the clean setting, the first paper receives 80% of "accept" in expectation and the second paper receives 90%. However, in the noisy setting, the first paper may still obtain 80% "accept" in expectation while the second paper may only obtain 60% "accept" in expectation because it receives more noisy ratings. In addition to the noise level, the noise can be systematically biased due to the existence of "cheap signals".

**Example 3** (Long proof vs. short proof). *The first paper has a complicated proof. The second paper has a short proof. In the noisy setting, each reviewer (unconsciously) intends to vote for "accept" for paper with a long proof, even if the length of the proof is a "cheap signal".*

It may occur in practice that the second paper is better in the clean setting, but the first paper receives more fraction of "accept" in expectation in the noisy setting, due to the **systematically biased** noise caused by cheap signals. Cheap signals can also be the authors' reputation, institutions, or writing skills. One may argue that the cheap signals may correlate with expensive signals, which represent the true quality of the paper. However, these cheap signals like proof length can be easily manipulated.

Prior knowledge and historical data may help to calibrate the noise. However, the identification of bias can be subjective and ad hoc. It will be more applicable if we do not need prior knowledge and design a general **one-shot** process. A fundamental question arises:

\**Can we design a one-shot scoring process that leads to a noise-robust rank without any prior knowledge? Formally, we want the paper with a higher expected score in the clean setting also have a higher score in the noisy setting with high probability, even if different papers' reviews have different noise levels and biases.*

A key challenge is that the ratings are subjective and unverifiable. As an outsider without any prior knowledge, we do not even know whether the bias exists or not. Therefore, it is impossible to address the above question without eliciting any additional information from the reviewers. Prelec [2] propose an innovative method, Surprisingly Popular (SP), to aggregate the information despite agents' systematic biases without any prior. Here each agent is asked to report both her signal (e.g. accept or reject) and her prediction (e.g. 90% reject, 10% accept) for a random peer's report. The elicited predictions are used to construct the prior. Instead of a majority vote, the aggregation is the signal which is more popular compared to the prior[2]. By assuming that the ground truth is positively correlated to each agent's signal[3], the aggregation is correct when the size of the crowds is sufficiently large.

We adopt the signal-prediction framework of SP, i.e., asking each reviewer to additionally provide her prediction for a randomly selected reviewer's report. Besides, we follow the literature of information elicitation [3] and information aggregation [4] to assume that agents are perfect Bayesian, which establishes a start for theoretical analysis:

**Assumption 1** (Bias generated from "cheap signals" can be predicted). *(Informal) We assume agents are perfect Bayesian. Besides, when their ratings are noisy and have systematic biases, they will adjust their predictions based on the biased noise.*

The above assumption essentially provides a general definition of the bias. With this assumption, we aim to design a one-shot scoring process with the reviewers' predictions as calibrations.

We cannot directly apply SP to this scenario, not only because it requires a large number of agents, but also because it is designed to solve a problem with objective ground truth. In the peer review setting, each paper's ratings are subjective and do not need to be the same even in the clean setting. We may ask the reviewer to compare two papers and apply SP to aggregate the comparisons. However, in practice, papers are reviewed independently by a possibly different set of reviewers.

Moreover, it is possible that both two papers' surprisingly popular ratings are "accept". To rank them, one idea is to compare the "amount" of surprise. For example, SP calibrates the ratings by dividing the prior, where the prior is constructed by the predictions. Subtracting the prior is another option. However, neither of the two calibration options is robust to the noise. In Example 2, when the prior is 50% / 50% for both the clean setting and the noisy setting, both of the two papers' surprisingly popular ratings are "accept". However, the second paper's "accept" is less popular compared to the prior because its ratings have more noise, while the second paper is better in the clean setting.

---

[2]In a 10% reject and 90% accept prior, 15% reject and 85% accept ratings make "reject" surprisingly popular.

[3]$\Pr[G = a | Y = a] > \Pr[G = a | Y = r]$, where $Y$ is a random agent's signal and $G$ denotes the ground truth.

We formulate the above problem into a formal mathematical model, defining a paper's true quality as its expected rating in the clean setting without noise, and assuming that the noise is non-degenerate and each reviewer's received noisy signal positively correlates with her clean signal[4].

**Our Results**   Within the model and assumptions, we propose a one-shot scoring process. When the number of reviewers is infinite, a paper with a higher score in the noisy setting will also have a higher true quality even if the papers have different biased noises. When the number of reviewers is finite, we provide a bound for the error probability and numerically show that our method beats the baseline, the average rating, in various situations. We also extend the results to non-binary setting naturally, with proper restrictions on the noise structures. Of an independent interest, we also provide surprisal measures that are invariant to noises.

**High-level Ideas**   The baseline score is the average rating, e.g., the fraction of "accept" votes. In this paper, we define the Surprisal-based Score in the form of $\frac{\text{baseline score} - \text{prior expected score}}{\text{correlation between ratings}}$, where the correlation is measured by a power of the determinant of the joint distribution matrix between two reviewers' ratings. Intuitively, the subtraction of the prior will cancel the effect of the systematic bias, and the division of the correlation will compensate for varying noise levels across contexts. For example, when faced with noisy reviewers providing weak signals beyond the prior, the average rating and the prior will closely align, resulting in a small correlation between ratings. By normalizing using this small correlation, the surprisal score is scaled up to compensate for the strong noise.

**Three Reviewers, Binary Ratings**   To illustrate our scoring process, we provide an example for the case of three reviewers, binary ratings.

**Example 4** (Three reviewers, binary ratings). *In the setting of binary ratings, the signal of reviewers is either "accept" (1) or "reject" (0). We call the reviewer who votes for "accept" as positive reviewer and the reviewer who votes for "reject" as negative reviewer. We map 3 "accept" signals to the highest score, and 3 "reject" signals to the lowest score. When there are at least an "accept" and a "reject", we construct a matrix $\hat{\mathbf{P}}$, where $\hat{P}_{s,t}$ is the average prediction, from the reviewers who report signal $s$, for the probability that a random reviewer votes for signal $t$. For example, $\hat{P}_{0,1}$ is the average of the negative reviewers' predictions for the probability that a random reviewer votes for "accept".*

*Specifically, the score is defined as*

$$\mathrm{S}(\cdot) = \begin{cases} +\infty & \textit{3 "accept", 0 "reject"} \\ \left(\frac{2}{3} - \hat{q}_1\right)\left(\hat{q}_1 \hat{q}_0 (\hat{P}_{1,1} - \hat{P}_{0,1})\right)^{-\frac{1}{2}} & \textit{2 "accept", 1 "reject"} \\ \left(\frac{1}{3} - \hat{q}_1\right)\left(\hat{q}_1 \hat{q}_0 (\hat{P}_{1,1} - \hat{P}_{0,1})\right)^{-\frac{1}{2}} & \textit{1 "accept", 2 "reject"} \\ -\infty & \textit{0 "accept", 3 "reject"} \end{cases}$$

*where $\hat{q}_1 = \frac{\hat{P}_{1,1}}{\hat{P}_{0,1} + \hat{P}_{1,0}}$ and $\hat{q}_0 = \frac{\hat{P}_{1,0}}{\hat{P}_{0,1} + \hat{P}_{1,0}}$.*

*Figure 1 illustrates how the score depends on the reviewers' predictions. At a high level, the score will be higher when the reviewers underestimate the probability that a random reviewer rates "accept" (the lower left corner)[5].*

To the best of our knowledge, we are the pioneer to rigorously formulate the noise-robust comparison problem with a one-shot scoring process, when different rating tasks have different noises. We believe our work provides a start in the peer review setting for calibrating the bias without any prior.

## 1.1   Related Work

**Bias in Peer Review**   Studies on peer review originated with Mahoney [5] and Hess [6]. Since then, numerous studies have emerged to identify sources and impacts of bias in peer review [7, 8, 9, 10, 11, 12, 13]. In addition to exploring sources and impacts, recent studies

---

[4]If not, the noisy reviewer can flip the rating sometimes. In this case, it is impossible to design a noise-robust scoring process without prior knowledge.

[5]In our model, in predicting the probability that a random reviewer votes for "accept", the positive reviewer will have a higher prediction than the negative reviewer. In practice, if this is not true, we will adopt the baseline score to rank the papers.

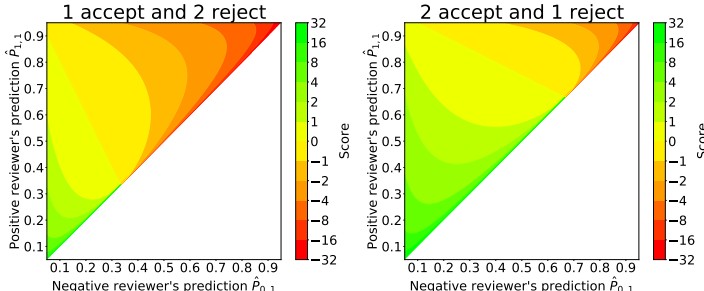

Figure 1: Contours of $S(\text{ratings},\text{predictions})$: We fix ratings (left: 1 accept, 2 reject; right: 1 reject, 2 accept) and illustrate the relationship between the score and the reviewers' predictions for the probability that a random reviewer votes for "accept".

have attempted to detect or calibrate biases automatically, employing various methods such as optimization, nonparametric estimation, and mechanism design [14, 15, 16, 17, 18, 19, 20]. However, these methods rely on historical data or a reviewer's votes for multiple papers.

**Mechanism Design via Second-order Information**   Prelec [3] introduced the signal-prediction framework where agents are asked to provide both their answers and predictions. By applying the signal-prediction framework, researchers have discovered methods to aggregate people's answers to the surprisingly popular option [2], reduce bias by agents' predictions about ranking [21], incentivise truthful agents[3, 22, 23], and find more informative answers [24, 25]. In addition to the discrete setting, recently, a growing literature [26, 27, 28, 29, 30, 31] has extended the signal-prediction framework to forecast aggregation settings. In this work, we employ second-order information for noise-robust comparison in a different setting.

## 2   Background and Problem Statement

### 2.1   Background: Single-task Information Elicitation

In this section, we describe how agents receive private signals and generate predictions, following the standard signal-prediction framework [2, 3].

**Model**   We regard each paper as a task with a *state* $\mathbf{w} \in [0,1]^{1 \times |\Sigma|}$, where $\Sigma$ is the set of all possible *signals*. There are $n$ *agents* (reviewers) assigned to the task[6], labeled from 1 to $n$. Each agent $i \in [n]$ receives private signal $X_i = s$ with probability $w_s$. The notation $x_i$ is the realization of $X_i$, representing the signal actually received by agent $i$. In this paper, we use the notation $s$ or $t$ to refer to a particular signal and the notation $i$ or $j$ to refer to a particular agent. There is a prior distribution $Q$ over the states and $W$ is a random state drawn from the prior distribution $Q$. We use a row vector $\mathbf{q} \in [0,1]^{1 \times |\Sigma|}$ to denote the prior probability that an agent will receive each signal, i.e., for all $s \in \Sigma$, $q_s = \Pr_Q[X_i = s] = \sum_{\mathbf{w}} \Pr_Q[W = \mathbf{w}] w_s$. Analogously, $\mathbf{P} \in [0,1]^{|\Sigma| \times |\Sigma|}$ denotes the prediction matrix such that $P_{s,t} = \Pr_Q[X_j = t | X_i = s], i \neq j$ is the probability that agent $j$ will receive the signal $t$ conditioning on another agent $i$ receives the signal $s$. The joint distribution matrix $\mathbf{U} \in [0,1]^{|\Sigma| \times |\Sigma|}$ is the matrix where $\forall s,t \in \Sigma, U_{s,t} = q_s P_{s,t} = \Pr_Q[X_i = s, X_j = t]$[7]. Because agents' identities do not matter, we omit $i,j \in [n]$ in the definition. To avoid degeneration, we assume the determinant of $\mathbf{U}$ is non-zero. In the task, each agent $i$ is asked to report her signal $s_i$ (e.g. accept) and her prediction $\mathbf{p}_i$ for a random agent's signal (e.g. 40% accept, 60% reject) without any communication to other agents.

The above model describes how agents receive signals **without noise**, which we call the clean setting. Figure 2 illustrates the model briefly and we offer a detailed example in Appendix A. We will introduce the noise model later.

---

[6] The number of reviewers can be flexible for different papers.

[7] The matrix $\mathbf{U}$ is always symmetric but $\mathbf{P}$ may not be symmetric.

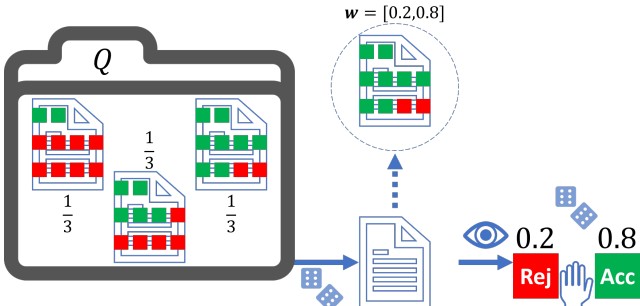

Figure 2: **Model without noise**: $Q$ is the prior distribution over the papers' states. In the figure, there are three types of states, each of which shows up with probability $\frac{1}{3}$. A random paper's state follows the distribution $Q$. When the paper's state realizes as $\mathbf{w} = [0.2, 0.8]$, a random reviewer of this paper will vote for "reject" with probability $0.2$ and vote for "accept" with probability $0.8$.

**Model Inference** For each agent $i \in [n]$, given her private signal $s_i$, we assume that her prediction $\mathbf{p}_i$ is the Bayesian posterior, i.e., $\forall t \in \Sigma, \mathbf{p}_i(t) = P_{s_i,t}$. If for all signal $s \in \Sigma$, there exists an agent who reports $s$, then we can obtain the prediction matrix $\mathbf{P}$ according to the agents' predictions. We can use $\mathbf{P}$ to induce the joint distribution matrix $\mathbf{U}$ where $\forall s, t \in \Sigma, U_{s,t} = q_s P_{s,t} = \Pr_Q[X_i = s, X_j = t]$.

**Claim 1** (Prelec et al. [2]). *We can construct the vector $\mathbf{q}$ and the joint distribution matrix $\mathbf{U}$ from the prediction matrix $\mathbf{P}$:* $\forall s, t \in \Sigma, q_s = (\sum_t \frac{P_{s,t}}{P_{t,s}})^{-1} (0/0 \equiv 0), U_{s,t} = q_s P_{s,t}$.

In this paper, we will assume the reviewers have good intentions and will tell the truth. That is, if she receives an "accept" signal for a paper, she will vote for "accept" as well. Regarding incentive design, prediction reports can be incentivized by proper scoring rules [32]. For the signal report, Prelec [3] develops a payment method called Bayesian Truth Serum (BTS) to incentivize agents to tell the truth when the number of agents is infinite. By assuming that agents perform the same strategy, Kong et al. [33] propose a method that only requires the number of agents greater or equal 6.

## 2.2 Problem Statement

**Noise Model** In the noisy setting, instead of observing the original signal $s$, which we call the *clean signal*, each agent can only observe its noisy version $M(s)$, which we call the *noisy signal*, where $M: \Sigma \mapsto \Sigma$ is a random operator. Denote $\mathbf{M}$ as the matrix where $M_{s,t}$ is the probability of observing noisy signal $t$ given clean signal $s$. We only consider the non-degenerate noise whose noise matrix $\mathbf{M}$ is invertible. Additionally, we focus on the homogeneous noise which corresponds to systematic biases. Homogeneous noise means that the noise is the same for all reviewers of a specific paper.

In the noisy setting, we use the notation $\hat{Q}$ to denote the prior distribution over the noisy states. Similar to the clean setting, $\hat{X}_i$ denotes the random variable of agent $i$'s noisy signal and $\hat{x}_i$ is its realization; $\hat{\mathbf{U}}$ denotes the joint distribution matrix where $\hat{U}_{s,t} = \Pr_{\hat{Q}}[\hat{X}_i = s, \hat{X}_j = t]$; $\hat{\mathbf{w}}$ denotes the noisy state; $\hat{\mathbf{q}}$ denotes the noisy prior over signals and $\hat{\mathbf{P}}$ denotes the noisy prediction matrix. Note that the relationship between $\hat{\mathbf{P}}, \hat{\mathbf{U}}$ and $\hat{\mathbf{q}}$ remains the same. By abusing notation a little bit, we sometimes write $\hat{Q} = M(Q), \hat{\mathbf{U}} = M(\mathbf{U}), \hat{\mathbf{w}} = M(\mathbf{w})$. Figure 3 briefly introduces how noise affects reviewers' ratings, and we offer a detailed example of noise model in Appendix A.

**Assumption 2.** *In the noisy setting, we assume that for each agent $i$, given her private noisy signal $s_i$, her prediction $\hat{\mathbf{p}}_i$ is still the Bayesian posterior where $\forall t \in \Sigma, \hat{\mathbf{p}}_i(t) = \Pr_{\hat{Q}}[X_j = t | X_i = s_i] = \hat{P}_{s_i,t}$.*

Ideally, we want to reconstruct the original state $\mathbf{w}$ from the noisy $\hat{\mathbf{w}}$ and $\hat{\mathbf{P}}$. However, we do not have a sufficient amount of information. Instead, we aim to solve the following problem.

**Noise-Robust Comparison** We have two rating tasks, say reviews for paper A and B, whose clean states are $\mathbf{w}_A$ and $\mathbf{w}_B$ respectively. Both $\mathbf{w}_A$ and $\mathbf{w}_B$ follow the distribution $Q$. There is a known one-to-one mapping $\varphi: \Sigma \to \mathbb{R}$ between signals in $\Sigma$ and paper scores. For example, when ratings are binary, the mapping $\varphi$ is $\{\text{reject} \to 0, \text{accept} \to 1\}$.

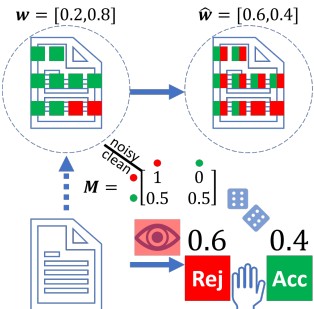
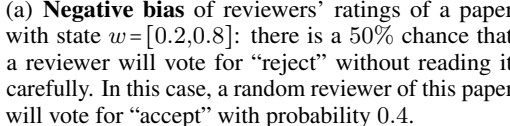
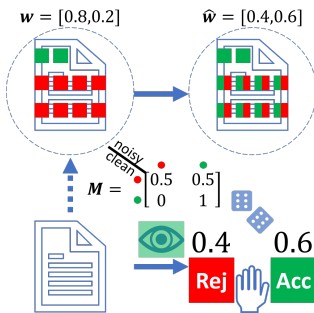

(a) **Negative bias** of reviewers' ratings of a paper with state $w = [0.2, 0.8]$: there is a 50% chance that a reviewer will vote for "reject" without reading it carefully. In this case, a random reviewer of this paper will vote for "accept" with probability 0.4.

(b) **Positive bias** of reviewers' ratings of a paper with state $w = [0.8, 0.2]$: there is a 50% chance that a reviewer will vote for "accept" without reading it carefully. In this case, a random reviewer of this paper will vote for "accept" with probability 0.6.

Figure 3: **Model with noise**

Therefore, for simplicity, we sometimes write signal $s$ as a score of real number and $\Sigma$ as a set of scores. For example, in the binary case, $\Sigma = \{0(\text{reject}), 1(\text{accept})\}$. We care about the *true quality* which is defined as the expected score in the clean state, i.e., $\mathbb{E}_{\mathbf{w}}\varphi = \sum_s \varphi(s)w_s$. For example, in the binary case, the quality is $\mathbb{E}_{\mathbf{w}}\varphi = w_1$, i.e., the probability of a random reviewer voting for "accept".

Paper A has $n_A$ reviewers and paper B has $n_B$ reviewers. They do not need to have the same number of reviewers. The pair $(\hat{x}_i^A, \hat{\mathbf{p}}_i^A)_{i \in [n_A]}$ denotes the reviewers' actual ratings and predictions for paper A, and $(\hat{x}_j^B, \hat{\mathbf{p}}_j^B)_{j \in [n_B]}$ analogously. Let $S(\cdot)$ be a function that takes reviewers' noisy ratings and predictions of a single paper as input, and outputs a calibrated score for the paper. For a pair of papers, we will rank the paper according to their calibrated scores[8]. Given a set of noises $\mathcal{M}$, we aim to design $S(\cdot)$ such that for all $M_A, M_B \in \mathcal{M}$, the *error probability* $\Pr[S(\cdot)$ ranks A higher than B|B's quality is better than A's] is upper-bounded and the upper bound goes to 0 when both $n_A$ and $n_B$ go to infinity.

## 3 Invariant Surprisal Vector

This section will introduce the key ingredient of our scoring process, a surprisal vector which is invariant to a natural family of noises. We first introduce this family of noises. Here each individual receives the clean signal with probability $1 - \lambda > 0$. With probability $\lambda$, which we call noise level, each individual receives a signal according to a distribution vector $\mathbf{b} \in \Delta_\Sigma$, which we call bias. We state the noise formally in the following definition.

**Definition 1** (A Family of Noises $\mathcal{M}^*$). *We consider a family of noises with two parameters, a noise level $\lambda \in [0, 1)$ and a bias vector $\mathbf{b} \in \Delta_\Sigma$, where $\Delta_\Sigma$ is the probability simplex on the signal set $\Sigma$. Let $\mathbf{B}$ denote the matrix whose rows are all $\mathbf{b}$. The noise is defined as $\mathbf{M}_{\lambda, \mathbf{b}} = (1 - \lambda)\mathbf{I} + \lambda\mathbf{B}$.*

**Claim 2.** *The noisy state $\hat{\mathbf{w}} = (1 - \lambda)\mathbf{w} + \lambda\mathbf{b}$ is a convex combination of the clean state $\mathbf{w}$ and the bias vector $\mathbf{b}$. In the binary case ($\Sigma = \{0(\text{reject}), 1(\text{accept})\}$), $\mathcal{M}^*$ is the set of all non-degenerate noises where $M_{1,1} > M_{0,1}$.*

Then, we define a surprisal vector that is invariant to the above noises.

**Definition 2** ($\mathcal{M}^*$-Invariant Surprisal Vector). *Given paper's state $\mathbf{w}$ and joint distribution $\mathbf{U}$, we define the $\mathcal{M}^*$-invariant surprisal vector as*

$$\text{Surp}^*(\mathbf{w}, \mathbf{U}) = \det(\mathbf{U})^{-\frac{1}{2(|\Sigma|-1)}}(\mathbf{w} - \mathbf{q}),$$

*where $\mathbf{q} \in [0, 1]^{1 \times |\Sigma|}$ denotes the marginal distribution of $\mathbf{U}$.*

---

[8]If two papers have the same score, we will randomly rank one higher, with a 50% chance for either.

| Notation | Description |
|---|---|
| $Q, \hat{Q}$ | $Q$ denotes the distribution over the clean states. $\hat{Q}$ denotes the distribution over the noisy states. |
| $\mathbf{w}, \hat{\mathbf{w}}$ | $\mathbf{w}$ denotes the paper's clean state. $\hat{\mathbf{w}}$ denotes the paper's noisy state. |
| $X_i, \hat{X}_i$ $x_i, \hat{x}_i$ | $X_i$ denotes a random variable indicating reviewer $i$'s clean signal for a paper which state follows $Q$. $\hat{X}_i$ denotes a random variable indicating reviewer $i$'s noisy signal for a paper which state follows $Q$. $x_i, \hat{x}_i$ are their realizations respectively. |
| $\mathbf{U}, \hat{\mathbf{U}}$ | $\mathbf{U}$ denotes the prior joint distribution over two reviewers' ratings, where $U_{s,t} = \Pr_Q[X_i = s, X_j = t]$. $\hat{\mathbf{U}}$ denotes its noisy version that $\hat{U}_{s,t} = \Pr_{\hat{Q}}[X_i = s, X_j = t]$. |
| $\mathbf{q}, \hat{\mathbf{q}}$ | $\mathbf{q}$ denotes the prior distribution vector over a reviewer's vote, where $q_s = \Pr_Q[X_i = s]$. $\hat{\mathbf{q}}$ denotes its noisy version that $\hat{q}_s = \Pr_{\hat{Q}}[X_i = s]$. |
| $\mathbf{P}, \hat{\mathbf{P}}$ | $\mathbf{P}$ denotes the prediction matrix, where $P_{s,t} = \Pr_Q[X_j = t|X_i = s], i \neq j$. $\hat{\mathbf{P}}$ denotes its noisy version where $\hat{P}_{s,t} = \Pr_{\hat{Q}}[X_j = t|X_i = s], i \neq j$. |
| $n$ | $n$ denotes the number of reviewers that are assigned for a paper. |
| $\mathbf{M}$ | $\mathbf{M}$ is the noise matrix defined by $\mathbf{M} = (1-\lambda)\mathbf{I} + \lambda \mathbf{1}^\top \mathbf{b}$, where $\lambda$ is the noise level and $\mathbf{b}$ is the bias vector. |
| $\mathbf{v}, \hat{\mathbf{v}}$ | The frequency vector of the ratings, where $v_s = \frac{1}{n}\sum_i \mathbf{1}[x_i = s]$. $\hat{\mathbf{v}}$ denotes its noisy version that $\hat{v}_s = \frac{1}{n}\sum_i \mathbf{1}[\hat{x}_i = s]$. |
| $\mathrm{S}^*(\cdot)$ | $\mathrm{S}^*(\cdot)$ is our Surprisal-based Score calculated from $\mathbf{w}$ and $\mathbf{U}$. |
| $\mathrm{S}(\cdot)$ | $\mathrm{S}(\cdot)$ is our Empirical Surprisal-based Score calculated from $n$ agents' ratings and predictions. |

Table 1: Notation table

**Claim 3.** *In the binary case ($\Sigma = \{0(reject), 1(accept)\}$), the $\mathcal{M}^*$-invariant surprisal vector can be simplified as*

$$\mathrm{Surp}^*(\mathbf{w}, \mathbf{U}) = \frac{\mathbf{w} - \mathbf{q}}{\sqrt{q_0 q_1 (P_{1,1} - P_{0,1})}}.$$

**Theorem 1** (Invariance). *The $\mathcal{M}^*$-invariant surprisal is non-negative, vanishes when the state $\mathbf{w}$ equals the prior $\mathbf{q}$, and is invariant to $\mathbf{M}_{\lambda, \mathbf{b}}$ for all $\lambda \in [0,1)$, $\mathbf{b} \in \Delta_\Sigma$. The implication of invariance is that for any noise with $\lambda \in [0,1)$, $\mathrm{Surp}^*(\mathbf{w}, \mathbf{U}) = \mathrm{Surp}^*(\hat{\mathbf{w}}, \hat{\mathbf{U}})$.*

The proofs of Claim 2, Claim 3 and Theorem 1 are deferred to Appendix C. For general noise, the direction of $\mathbf{w} - \mathbf{q}$ may not be invariant. For example, one possible noise is flipping the ratings, i.e., rating "accept" when "reject" and rating "reject" when "accept". In this case, the direction of $\mathbf{w} - \mathbf{q}$ will be flipped as well. However, if we only care about the amount of surprisal, we successfully design a measure that is invariant to all possible non-degenerate noise and show that it can be used to identify the true state $\mathbf{w}$ in some scenarios (see Appendix B).

## 4 Surprisal-based Score

This section introduces our scoring process based on the surprisal vector introduced in Section 3. We first introduce the score based on "wishful thinking", i.e., when we have the state (or the noisy state). We then resolve this "wishful thinking" by providing an estimation of the state.

**Definition 3** (Surprisal-based Score). *Given $\mathbf{w}$ and $\mathbf{U}$, we define the $\mathcal{M}^*$-invariant Surprisal-based Score as*

$$\mathrm{S}^*(\mathbf{w}, \mathbf{U}) = (\mathbb{E}_{\mathbf{w}}\varphi - \mathbb{E}_{\mathbf{q}}\varphi)\det(\mathbf{U})^{-\frac{1}{2(|\Sigma|-1)}}.$$

*According to the simplification in the binary case (Claim 3), the $\mathcal{M}^*$-invariant surprisal score in the binary case can be simplified to*

$$\mathrm{S}^*(\mathbf{w}, \mathbf{U}) = \frac{w_1 - q_1}{\sqrt{q_0 q_1 (P_{1,1} - P_{0,1})}}.$$

Theorem 1 directly implies the following results: given the noisy states, we can compare their Surprisal-based Scores $\mathrm{S}^*(\mathbf{w}, \mathbf{U})$ and the comparison is invariant to the noise in $\mathcal{M}^*$ and consistent to the comparison of their true qualities.

**Corollary 1** (Noise-invariant comparison by $S^*(\cdot)$). *For any two states $\mathbf{w}_A, \mathbf{w}_B$ which follow distribution Q, and any two noises $M_A, M_B \in \mathcal{M}^*$,*

$$S^*(M_A(\mathbf{w}_A), M_A(\mathbf{U})) - S^*(M_B(\mathbf{w}_B), M_B(\mathbf{U}))$$
$$= S^*(\mathbf{w}_A, \mathbf{U}) - S^*(\mathbf{w}_B, \mathbf{U})$$
$$\propto \mathbb{E}_{\mathbf{w}_A}\varphi - \mathbb{E}_{\mathbf{w}_B}\varphi.$$

When there are only a small number of reviewers, we will use an estimation of the Surprisal-based Score, which we call the Empirical Surprisal-based Score. We first focus on the binary case. We have already proved that for any invertible noise matrix $\mathbf{M}$ where $M_{1,1} > M_{0,1}$, the metric $\frac{w_1 - q_1}{\sqrt{(q_0 q_1 (P_{1,1} - P_{0,1}))}}$ can be used to implement a noise-invariant comparison (Claim 2, Corollary 1). We will use the reviewers' ratings and predictions to estimate the score. We employ a *frequency vector* $\hat{\mathbf{v}} \in [0,1]^{1 \times |\Sigma|}$ to denote the frequency of the reviewers' reported signals. Formally, for all $s \in \Sigma$, $\hat{v}_s = \frac{1}{n}\sum_{i=1}^n \mathbf{1}[\hat{x}_i = s]$, where $\hat{x}_i$ is the reported noisy signal of reviewer $i$. For example, in the binary case, $\hat{\mathbf{v}} = (\hat{v}_0, \hat{v}_1)$ where $\hat{v}_1$ is the fraction of "accept" and $\hat{v}_0$ is the fraction of "reject".

**Empirical Surprisal-based Score (Binary)** When there exists at least one "accept" and one "reject", we construct a $2 \times 2$ prediction matrix $\hat{\mathbf{P}}$ based on the reviewers' ratings and predictions. Each element of the matrix, $\hat{P}_{s,t}$, is the average prediction from the reviewers who report the signal $s$, for the probability that a random reviewer reports the signal $t$. For example, $\hat{P}_{0,1}$ is the average of the negative reviewers' prediction for the probability a random reviewer votes for "accept". When reviewers' ratings are the same, the matrix $\mathbf{P}$ cannot be constructed. In this case, we set $+\infty$ score for the papers that all reviewers vote for "accept" and $-\infty$ for the papers that all reviewers vote for "reject". Formally,

$$S((\hat{x}_i, \hat{\mathbf{p}}_i)_{i \in [n]}) = \begin{cases} -\infty & \hat{v}_1 = 0 \\ \frac{\hat{v}_1 - \hat{q}_1}{\sqrt{\hat{q}_0 \hat{q}_1 (\hat{P}_{1,1} - \hat{P}_{0,1})}} & \hat{v}_1 \in (0,1) \\ +\infty & \hat{v}_1 = 1 \end{cases},$$

where $\hat{q}_1 = \frac{\hat{P}_{1,1}}{\hat{P}_{0,1} + \hat{P}_{1,0}}$ and $\hat{q}_0 = \frac{\hat{P}_{1,0}}{\hat{P}_{0,1} + \hat{P}_{1,0}}$ (See Claim 1 for the calculation of $\hat{\mathbf{q}}$ from $\hat{\mathbf{P}}$).

We naturally extend the process to non-binary settings. For example, review signals contain multiple grades ($\Sigma = \{-2(\text{reject}), -1(\text{weak reject}), 1(\text{weak accept}), 2(\text{accept})\}$).

**Empirical Surprisal-based Score (General)** When $\hat{v}_s > 0\ \forall s \in \Sigma$, we construct the prediction matrix $\hat{\mathbf{P}}$. The calibrated score is defined as

$$S((\hat{x}_i, \hat{\mathbf{p}}_i)_{i \in [n]}) = (\mathbb{E}_{\hat{\mathbf{v}}}\varphi - \mathbb{E}_{\hat{\mathbf{q}}}\varphi)\det(\hat{\mathbf{U}})^{-\frac{1}{2(|\Sigma|-1)}},$$

where $\forall s \in \Sigma, \hat{q}_s = (\sum_t \frac{\hat{P}_{s,t}}{\hat{P}_{t,s}})^{-1}$ ($0/0 \equiv 0$). When $\det(\hat{\mathbf{U}}) < 0$, $S(\cdot)$ remains undefined. This "bad event" arises when a reviewer favors a paper, her belief in the likelihood of another reviewer also favoring it decreases. This unusual scenario alerts the Program Committee that this particular paper warrants further discussion and careful consideration before making a final decision. When comparing tasks with undefined scores, our method degenerates to the baseline, i.e., comparing the value of $\mathbb{E}_{\hat{\mathbf{v}}}\varphi$.

## 5 Theoretical Guarantee

In this section, we theoretically analyse the performance of our Empirical Surprisal-based Score. We use the *error probability* to measure the performance. Recall that we use the expected score of a paper in the clean state to measure its true quality. In the binary case, the true quality of a paper, whose clean state is $\mathbf{w}$, is $w_1$, i.e., the fraction of "accept" ratings in the clean setting. The error probability is the probability that the score ranks paper A higher than paper B while B's quality is better than A's.

Theorem 2 (binary) and Theorem 3 (general) show the theoretical upper bound of the error probability of our method. When the number of reviewers goes to infinity, the error probability goes to zero. The analysis follows from the results of Corollary 1 and a standard concentration bound analysis. We defer the proofs of these two theorems to Appendix C.

**Theorem 2** (Error probability, binary case). *For any pair of papers $A, B$ whose clean states are $\mathbf{w}^A, \mathbf{w}^B$ correspondingly (without loss of generality, let $w_1^A < w_1^B$), given their noise matrices*

$\mathbf{M}^A, \mathbf{M}^B \in \mathcal{M}^*$ *correspondingly, the error probability* $\Pr[\mathrm{S}(A) > \mathrm{S}(B)|w_1^A, w_1^B] + \frac{1}{2}\Pr[\mathrm{S}(A) = \mathrm{S}(B)|w_1^A, w_1^B]$ *goes to 0 when the number of reviewers* $n_A, n_B$ *goes to infinity, and is bounded by*

$$\left(\hat{w}_1^B\right)^{n_A} + \left(1 - \hat{w}_1^B\right)^{n_B} - \frac{1}{2}\left[\left(\hat{w}_1^A\right)^{n_A}\left(\hat{w}_1^B\right)^{n_B} + \left(1 - \hat{w}_1^A\right)^{n_A}\left(1 - \hat{w}_1^B\right)^{n_B}\right] + \exp\left\{-\frac{2(w_1^B - w_1^A)^2}{\frac{1}{n_A(1-\lambda_A)^2} + \frac{1}{n_B(1-\lambda_B)^2}}\right\},$$

*where $\lambda_A$ is the noise level of paper A and $\lambda_B$ is the noise level of paper B*[9].

The error bound has three terms. The first two terms bound the error probability conditioning on at least one of the papers having an infinite score, i.e., every reviewer votes for "accept" ("reject"). The last term bounds the error probability conditioning on both papers has at least an "accept" and a "reject" vote. This theorem shows that we can achieve a better error bound when the rating difference $(w_1^B - w_1^A)$ between two papers is larger, when there are more reviewers ($n_A$ and $n_B$), and when the noise levels ($\lambda_A$ and $\lambda_B$) are lower.

**Theorem 3** (Error probability, general case). *For any pair of tasks $A, B$ whose clean states are* $\mathbf{w}^A, \mathbf{w}^B$ *correspondingly (without loss of generality, let $\mathbb{E}_{\mathbf{w}^A}\varphi < \mathbb{E}_{\mathbf{w}^B}\varphi$), given their noise matrices* $\mathbf{M}^A, \mathbf{M}^B \in \mathcal{M}^*$ *correspondingly, the error probability* $\Pr[\mathrm{S}(A) > \mathrm{S}(B)|\mathbf{w}^A, \mathbf{w}^B] + \frac{1}{2}\Pr[\mathrm{S}(A) = \mathrm{S}(B)|\mathbf{w}^A, \mathbf{w}^B]$ *goes to 0 when the number of reviewers $n_A, n_B$ goes to infinity, and is bounded by*[10]

$$\sum_{s \in \Sigma}(1 - \hat{w}_s^A)^{n_A} + \sum_{s \in \Sigma}(1 - \hat{w}_s^B)^{n_B} + \exp\left\{-\frac{2(\mathbb{E}_{\mathbf{w}^B}\varphi - \mathbb{E}_{\mathbf{w}^A}\varphi)^2}{(\varphi_{\max} - \varphi_{\min})^2\left(\frac{1}{n_A(1-\lambda_A)^2} + \frac{1}{n_B(1-\lambda_B)^2}\right)}\right\}.$$

## 6 Numerical Experiments

We perform numerical experiments to compare the performance of our Surprisal-based Score and the baseline score in the binary setting ($\Sigma = \{0(\text{reject}), 1(\text{accept})\}$). Recall that the baseline is the proportion of the "accept" ratings. Here we describe the parameters we select in numerical experiments.

1. **The number of agents** $n$: We perform the experiments in the settings of $n = 3$ and $n = 5$.

2. **The prior distribution of states** $Q$: In the binary case, there exists a one-to-one mapping between the state $\mathbf{w} = (w_0 = 1 - w_1, w_1)$ and $w_1 \in [0, 1]$, which is the probability that a random reviewer votes for "accept" in the clean setting. The prior over the states can be described by a prior distribution over $w_1$. We use three different priors for $w_1$, which are $\text{Beta}(\frac{1}{2}, \frac{1}{2})$(most papers' quality is either high or low), $\text{Beta}(1, 1)$(papers' quality is distributed uniformly), and $\text{Beta}(3, 3)$(most papers have a medium quality).

3. **The bias vector** $\mathbf{b}$: Since the numerical experiment is to compare the scores of two papers, we consider the opposite and same biases between two papers:

   - **opposite**: Paper A has the positive bias vector $\mathbf{b}_A = [0, 1]$ where the reviewers intend to vote for "accept" without careful review. Paper B has an negative bias vector $\mathbf{b}_B = [1, 0]$. This simulates a situation where one paper has a negative cheap signal and another paper has a positive cheap signal.

   - **same**: Both paper A and B have the same bias vector $\mathbf{b}_A = \mathbf{b}_B = [0, 1]$[11]. This simulates a situation where both papers have positive (negative) cheap signals.

We evaluate the performance by accuracy, which is 1 minus the expected error probability, and perform the experiments when two papers have opposite biases (Figure 4) and the same bias (Figure 5). In each setting, we consider three scenarios with varying noise levels for paper A ($\lambda_A = 0, 0.3, 0.6$). In each scenario, we vary the noise level of paper B and compare our Surprisal-based Score and baseline under different priors and the number of reviewers. The x axis is paper B's noise level and the y axis is the accuracy of Surprisal-based Score (red lines) and baseline (green lines)[12]. According to the results, our method outperforms or equals the baseline in accuracy in all cases. We also notice that our method has more advantages under the opposite bias setting and when the number of reviewers is higher.

---

[9]Note that the bound is independent of the biases.

[10]$\varphi_{\max}$ and $\varphi_{\min}$ are the abbreviations of $\max_{s \in \Sigma}\varphi(s)$ and $\min_{s \in \Sigma}\varphi(s)$ respectively.

[11]Since the prior we choose is symmetric, for the case of $\mathbf{b}_A = \mathbf{b}_B = [1, 0]$, the result is exactly the same.

[12]The discontinuity happens because when $\lambda_A$ is fixed and $\lambda_B$ increases to a certain threshold, 1 "accept's" score can be higher than 2 "accept's" score.

In addition, we propose another benchmark for comparison, named the SP-inspired score. This benchmark slightly modifies the original Surprisingly Popular method [2] to fit our setting. This benchmark is not invariant to noise but usually has a lower variance. We include the details and the numerical experiments comparing our method with the SP-inspired score in Appendix D.

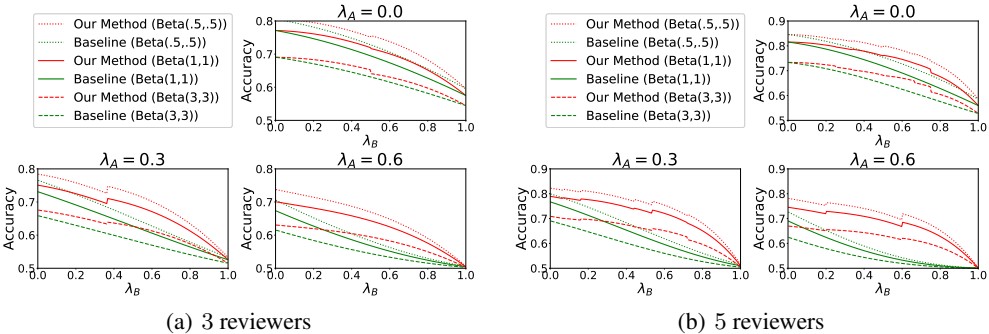

Figure 4: **Performance evaluation (opposite biases)**

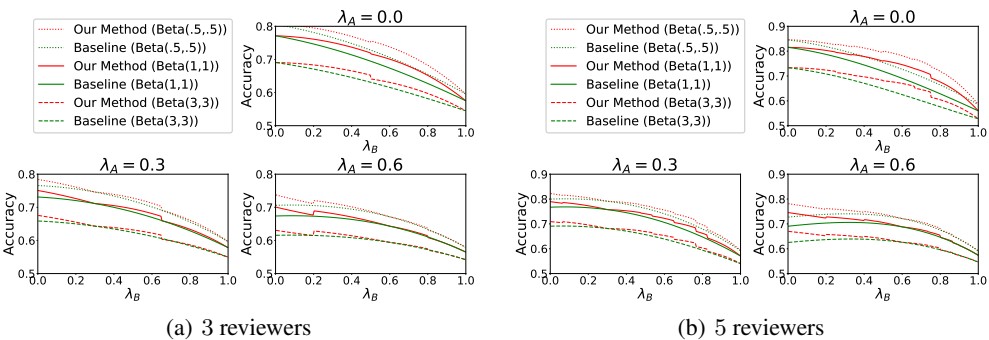

Figure 5: **Performance evaluation (same bias)**

# 7 Conclusion and Discussion

When papers receive ratings of different noises due to heterogeneity of paper topics and the existence of cheap signals, we propose a scoring process that leads to a noise-robust rank among papers. The implementation of the process requires the reviewers to report their predictions for other ratings but does not require any prior knowledge. We provide theoretical and numerical justification for the process.

Our theory assumes that the reviewers are perfect Bayesian. However, this may not be the case in practice. Non-Bayesian reviewers may exhibit reporting bias and prediction bias. To some extent, prediction bias is inevitable even with a rational crowd, since it is hard for reviewers to fully understand the prior. The good news is, if the prediction bias is identically distributed among reviewers, then it can be regarded as part of the noisy prior, making the solution we propose still feasible. In contrast, dealing with reporting bias in the one-shot setting appears to be more challenging, since it can violet the basic assumption of homogeneous reviewers. For instance, some reviewers may be more lenient, while others are more strict. Many related works have considered this situation and proposed solutions to mitigate or calibrate the reporting error. We can employ any of these report debiasing schemes based on historical information or multi-tasking, and then use our Surprisal-based Score on the processed reports to measure the paper quality.

# 8 Acknowledgements

This research is supported by National Key R&D Program of China (2022ZD0114900).

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

# A Detailed Examples of Our Model

**Example 5** (Binary case, clean setting). *Signal $0$ represents "reject" and signal $1$ represents "accept". There are three types of papers, $\alpha$, $\beta$, and $\gamma$. Type $\alpha$ paper's state is $\mathbf{w}^\alpha = [w_0^\alpha, w_1^\alpha] = [0.8, 0.2]$. That is, in the clean setting, a reviewer will receive "accept" signal for type $\alpha$ paper with probability $w_1^\alpha = 0.2$. Analogously, we let $w_1^\beta = 0.5$ and $w_1^\gamma = 0.8$. A random paper's state has type $\alpha$, $\beta$, and $\gamma$ with equal probability. This describes the prior distribution over paper's states Q:*

$$\mathbf{w} = \begin{cases} \mathbf{w}^\alpha \ w.p. \ \frac{1}{3} \\ \mathbf{w}^\beta \ w.p. \ \frac{1}{3} \\ \mathbf{w}^\gamma \ w.p. \ \frac{1}{3} \end{cases}.$$

*The prior probability that a reviewer receive "reject" signal for a random paper is $q_0 = \frac{1}{3}w_0^\alpha + \frac{1}{3}w_0^\beta + \frac{1}{3}w_0^\gamma = 0.5, q_1 = 1 - q_0 = 0.5$. The probability that both of the two reviewers receive "reject" for a paper will be $U_{0,0} = \frac{1}{3}(w_0^\alpha)^2 + \frac{1}{3}(w_0^\beta)^2 + \frac{1}{3}(w_0^\gamma)^2 = 0.31$. In general, the joint distribution between two reviewers' ratings is represented as a matrix:*

$$\begin{aligned} \mathbf{U} &= \frac{1}{3}\mathbf{U}^\alpha + \frac{1}{3}\mathbf{U}^\beta + \frac{1}{3}\mathbf{U}^\gamma \\ &= \frac{1}{3}\begin{bmatrix} w_0^\alpha \\ w_1^\alpha \end{bmatrix}[w_0^\alpha, w_1^\alpha] + \frac{1}{3}\begin{bmatrix} w_0^\beta \\ w_1^\beta \end{bmatrix}[w_0^\beta, w_1^\beta] + \frac{1}{3}\begin{bmatrix} w_0^\gamma \\ w_1^\gamma \end{bmatrix}[w_0^\gamma, w_1^\gamma] \\ &= \frac{1}{3}\begin{bmatrix} 0.8 \\ 0.2 \end{bmatrix}[0.8, 0.2] + \frac{1}{3}\begin{bmatrix} 0.5 \\ 0.5 \end{bmatrix}[0.5, 0.5] + \frac{1}{3}\begin{bmatrix} 0.2 \\ 0.8 \end{bmatrix}[0.2, 0.8] \\ &= \begin{bmatrix} 0.31 & 0.19 \\ 0.19 & 0.31 \end{bmatrix}. \end{aligned}$$

*According to the joint distribution matrix $\mathbf{U}$, the prediction matrix $\mathbf{P}$ can be established:*

$$\mathbf{P} = \begin{bmatrix} \frac{U_{0,0}}{U_{0,0}+U_{0,1}} & \frac{U_{0,1}}{U_{0,0}+U_{0,1}} \\ \frac{U_{1,0}}{U_{1,0}+U_{1,1}} & \frac{U_{1,1}}{U_{1,0}+U_{1,1}} \end{bmatrix} = \begin{bmatrix} 0.62 & 0.38 \\ 0.38 & 0.62 \end{bmatrix}.$$

*Thus, when a reviewer receives "reject" signal, she will believe a random reviewer will receive "accept" signal with probability $P_{0,1} = 0.38$. When a reviewer receives "accept" signal, she will believe a random reviewer will receive "accept" signal with probability $P_{1,1} = 0.62$.*

**Example 6** (Binary case, noisy setting). *We use the same prior in Example 5. Given a paper of type $\gamma$ (i.e. the probability that the clean signal of a reviewer is "accept" is $0.8$), the paper is difficult to read so that there is a $50\%$ chance that a reviewer will vote for "reject" without reading it carefully. This shows that the paper has a noise matrix $\mathbf{M} = \begin{bmatrix} 1 & 0 \\ 0.5 & 0.5 \end{bmatrix}$. When the noise applies, $\forall s, t \in \Sigma, \hat{U}_{s,t} = \sum_{a,b} U_{a,b} M_{a,s} M_{b,t}$. Thus, $\hat{\mathbf{U}} = \mathbf{M}^\top \mathbf{U} \mathbf{M}$, thereby we obtain*

$$\hat{\mathbf{U}} = \mathbf{M}^\top \mathbf{U} \mathbf{M} \approx \begin{bmatrix} 0.58 & 0.17 \\ 0.17 & 0.08 \end{bmatrix},$$

$$\hat{\mathbf{P}} = \begin{bmatrix} \frac{\hat{U}_{0,0}}{\hat{U}_{0,0}+\hat{U}_{0,1}} & \frac{\hat{U}_{0,1}}{\hat{U}_{0,0}+\hat{U}_{0,1}} \\ \frac{\hat{U}_{1,0}}{\hat{U}_{1,0}+\hat{U}_{1,1}} & \frac{\hat{U}_{1,1}}{\hat{U}_{1,0}+\hat{U}_{1,1}} \end{bmatrix} \approx \begin{bmatrix} 0.77 & 0.23 \\ 0.68 & 0.32 \end{bmatrix}.$$

*If a reviewer receives and votes for "reject", she will predict a random reviewer will receive and vote for "accept" with probability $\hat{P}_{0,1} = 0.23$. If a reviewer receives and votes for "accept", she will predict a random reviewer will receive and vote for "accept" with probability $\hat{P}_{1,1} = 0.32$.*

# B Invariant Surprisal

We introduce a measure that is invariant to any invertible noise.

**Definition 4** (Invariant Surprisal). *Given state $\mathbf{w}$ and joint distribution $\mathbf{U}$, we define the invariant surprisal as*

$$\mathrm{Surp}(\mathbf{w}, \mathbf{U}) = (\mathbf{w} - \mathbf{q})\mathbf{U}^{-1}(\mathbf{w}^\top - \mathbf{q}^\top)$$

*where prior $\mathbf{q}$ is the marginal distribution of $\mathbf{U}$.*

The invariant surprisal is closely related to the affine-invariant Mahalanobis distance, which measures the distance between a point $\mathbf{x}$ and a distribution $D$ by $d_M(\mathbf{x}, D) = \sqrt{(\mathbf{x} - \boldsymbol{\mu})\mathbf{S}^{-1}(\mathbf{x} - \boldsymbol{\mu})}$ where

$\mu = \mathbb{E}[\mathbf{x}]$ and $\mathbf{S}$ is the covariance matrix. The invariant surprisal is closely related to $d_M(\mathbf{w},Q)^2$. However, we cannot directly use $d_M(\mathbf{w},Q)^2$. In our setting, $\mathbf{w}$ is a distribution vector where the sum of the elements is 1, thus the covariance matrix $\mathbf{S}$ is degenerate[13] and $\mathbf{S}^{-1}$ is not well-defined. Thus, we modify $d_M(\mathbf{w},Q)^2$ by using the simple joint distribution matrix $\mathbf{U}$ rather than the covariance matrix. It is invariant and well-defined. When the number of distinct states is $|\Sigma|$, it matches Shannon's surprisal to some extent.

**Theorem 4.** *The invariant surprisal is non-negative, vanishes when the state $\mathbf{w} = \mathbf{q}$ equals the prior, and is invariant to any invertible noise. In the binary case where $\Sigma = \{0,1\}$, the invariant surprisal is* $\mathrm{Surp}(\mathbf{w}, \mathbf{U}) = (\det(\mathbf{U}))^{-1}(w_1 - q_1)^2 = (q_0 q_1(P_{1,1} - P_{0,1}))^{-1}(w_1 - q_1)^2$ *and* $q_0 = \frac{P_{1,0}}{P_{0,1}+P_{1,0}}$, $q_1 = \frac{P_{0,1}}{P_{0,1}+P_{1,0}}$.

*Proof of Theorem 4.*

$$(\hat{\mathbf{w}}-\hat{\mathbf{q}})\hat{\mathbf{U}}^{-1}(\hat{\mathbf{w}}-\hat{\mathbf{q}})^\top$$
$$=(\mathbf{w}-\mathbf{q})\mathbf{M}(\mathbf{M}^\top\mathbf{U}\mathbf{M})^{-1}\mathbf{M}^\top(\mathbf{w}-\mathbf{q})^\top$$
$$=(\mathbf{w}-\mathbf{q})\mathbf{U}^{-1}(\mathbf{w}-\mathbf{q})^\top$$

In the binary case where $\Sigma = \{\text{reject},\text{accept}\}$, recall that we use 1 for accept and 0 for reject for short. Then, we can simplify the expression of surprisal in the binary case as follow.

$$\mathrm{Surp}(\mathbf{w},\mathbf{U})$$
$$=(\mathbf{w}-\mathbf{q})\mathbf{U}^{-1}(\mathbf{w}^\top-\mathbf{q}^\top)$$
$$=\begin{bmatrix}w_0-q_0 & w_1-q_1\end{bmatrix}\frac{1}{\det(\mathbf{U})}\begin{bmatrix}U_{1,1} & -U_{0,1}\\ -U_{1,0} & U_{0,0}\end{bmatrix}\begin{bmatrix}w_0-q_0\\ w_1-q_1\end{bmatrix}$$
$$=\frac{1}{\det(\mathbf{U})}(w_0-q_0)^2(\sum_{s,t\in\Sigma}U_{s,t}) \qquad\qquad (w_0-q_0=-(w_1-q_1))$$
$$=\frac{1}{\det(\mathbf{U})}(w_0-q_0)^2$$
$$=(q_0 q_1(P_{0,0}-P_{1,0}))^{-1}(w_0-q_0)^2$$
$$=(q_0 q_1(P_{1,1}-P_{0,1}))^{-1}(w_1-q_1)^2$$

$\square$

The invariant surprisal is invariant to any invertible noise. However, it only reflects the distance between the state and the prior and is not suitable for many application scenarios like peer review. Nevertheless, we show that when people reach a consensus in the clean state, the invariant surprisal can be used to identify the clean state. Moreover, in this case, the invariant surprisal matches Shannon's surprisal to some degree.

**Shannon's surprisal**  For a random variable $X$ where $\forall x, \Pr[X = x] = q_x$, Shannon's definition of the surprisal score of the outcome $x$ is $\log\frac{1}{q_x}$. When the outcome is more likely to happen, the surprisal that it actually happens is small. To measure the surprisal of a state $\mathbf{w}$, we can use $\log\frac{1}{q_{\mathbf{w}}}$ where $q_{\mathbf{w}}$ denotes the probability that the state $\mathbf{w}$ shows up.

**Example 7** (The consensus case: Invariant Surprisal $\approx$ Surprisal)**.** *We consider a special case, call it the consensus case, where agents always have the same rating. Formally there are $|\Sigma|$ distinct states and each state is a one-hot vector. For all $s$, the $s^{th}$ state's coordinates are all zero except that the $s^{th}$ coordinate is 1, denoted by $\mathbf{1}_s$. For example, when $d=3$, the states are $(1,0,0),(0,1,0),(0,0,1)$. In this case, $q_s$ is not only the prior probability for signal $s$, but also the probability that the $s^{th}$ state shows up. Let $\mathrm{diag}(\mathbf{q})$ denote a diagonal matrix whose diagonal elements consist of vector*

---

[13]For example, $\mathbf{w}$ is either $(1,0)$ or $(0,1)$ with equal probability. $\mathbf{S} = \begin{bmatrix} 0.25 & -0.25 \\ -0.25 & 0.25 \end{bmatrix}$

**q**. $\mathbf{U} = \mathrm{diag}(\mathbf{q})$. *The invariant surprisal for the $s^{th}$ state is*

$$\mathrm{Surp}(\mathbf{1}_s, \mathbf{U})$$
$$= (\mathbf{1}_s - \mathbf{q})\mathrm{diag}(\mathbf{q})^{-1}(\mathbf{1}_s^\top - \mathbf{q}^\top)$$
$$= \frac{1}{q_s} - 2 + 1$$
$$= \frac{1}{q_s} - 1.$$

In the consensus case, there is a one-to-one correspondence between states and signals. Our definition for the surprisal of the state $\mathbf{1}_s$ where all agents agree to signal $s$ is $\frac{1}{q_s} - 1 = \frac{1}{q_{\mathbf{1}_s}} - 1$ which is the 1st Taylor expansion of Shannon's.

In a special case where there are only $|\Sigma|$ distinct states, we have the following result.

**Corollary 2** (The noisy consensus case). *When there are only $|\Sigma|$ distinct states $\mathbf{w}_1, \mathbf{w}_2, \cdots, \mathbf{w}_{|\Sigma|}$, the invariant surprisal of each state $\mathbf{w}_k$ is the 1st Taylor expansion of Shannon's surprisal, i.e.,* $\mathrm{Surp}(\mathbf{w}_k, \mathbf{P}) = \frac{1}{q_{\mathbf{w}_k}} - 1$.

We will show that the above case is a noisy version of the consensus case. The invariance directly induces the above result.

*Proof of Corollary 2.* When there are only $|\Sigma|$ distinct states, we can define the intrinsic noise $\mathbf{M}$ as a matrix where each $k^{th}$ row is $\mathbf{w}_k$ such that it is the noisy version of the consensus case. Then the invariance of the invariant surprisal leads to the above corollary. $\qquad\square$

**Identifying Clean State by the Invariant Surprisal**   Given that agents reach consensus in the clean states, if we have already known the prior $\mathbf{q}$ and the prior is unbalanced $q_s \neq q_t, \forall s \neq t$, then we can identify the clean state in the noisy setting by the invariant surprisal. Sometimes even side information about $\mathbf{q}$ is sufficient. For example, in the binary case, if we know that the fraction of all "accept" is less than .5, i.e., $q_a < 0.5$, then once we obtain $> 1$ invariant surprisal in the noisy setting, we know the clean state is all "accept", otherwise, the clean state is all "reject".

## C   Omitted Proofs

### C.1   Proof of Claim 2

**Claim 2.** *The noisy state $\hat{\mathbf{w}} = (1 - \lambda)\mathbf{w} + \lambda\mathbf{b}$ is a convex combination of the clean state $\mathbf{w}$ and the bias vector $\mathbf{b}$. In the binary case ($\Sigma = \{0(reject), 1(accept)\}$), $\mathcal{M}^*$ is the set of all non-degenerate noises where $M_{1,1} > M_{0,1}$.*

*Proof of Claim 2.* First, we show that $\hat{\mathbf{w}} = (1 - \lambda)\mathbf{w} + \lambda\mathbf{b}$.

$$\hat{\mathbf{w}} = \mathbf{w}\mathbf{M}_{\lambda, \mathbf{b}} \qquad\qquad\qquad\qquad \text{(see Claim 5 for more details)}$$
$$= \mathbf{w}((1 - \lambda)\mathbf{I} + \lambda\mathbf{B}) \qquad\qquad\qquad \text{(Definition of } \mathbf{M}_{\lambda, \mathbf{b}})$$
$$= (1 - \lambda)\mathbf{w} + \lambda\mathbf{b}$$

Then, we prove that every binary $\mathbf{M} = \mathbf{M}_{\lambda, \mathbf{b}} \in \mathcal{M}^*$ is invertible and satisfies $M_{1,1} > M_{0,1}$.

$$\mathbf{M}_{\lambda, \mathbf{b}} = (1 - \lambda)\mathbf{I} + \lambda\mathbf{B} = \begin{bmatrix} 1 - \lambda + \lambda b_0 & \lambda b_1 \\ \lambda b_0 & 1 - \lambda + \lambda b_1 \end{bmatrix}$$
$$\det(M) = M_{0,0}M_{1,1} - M_{0,1}M_{1,0} = 1 - \lambda \neq 0$$
$$M_{1,1} - M_{0,1} = 1 - \lambda > 0$$

Finally, we prove that every invertible binary noise matrix $\mathbf{M}$ with $M_{1,1} > M_{0,1}$ belongs to $\mathcal{M}^*$.

$$\mathbf{M} = \begin{bmatrix} 1 - M_{0,1} & M_{0,1} \\ 1 - M_{1,1} & M_{1,1} \end{bmatrix}$$

$$= (M_{1,1} - M_{0,1})\mathbf{I} + (1 - M_{1,1} + M_{0,1}) \begin{bmatrix} 1 - \frac{M_{0,1}}{1 - M_{1,1} + M_{0,1}} & \frac{M_{0,1}}{1 - M_{1,1} + M_{0,1}} \\ 1 - \frac{M_{0,1}}{1 - M_{1,1} + M_{0,1}} & \frac{M_{0,1}}{1 - M_{1,1} + M_{0,1}} \end{bmatrix}$$

Let $\lambda = 1 - M_{1,1} + M_{0,1}$ and $\mathbf{b} = \left[ 1 - \frac{M_{0,1}}{\lambda}, \frac{M_{0,1}}{\lambda} \right]$. Then, we have $\mathbf{M} = (1 - \lambda)\mathbf{I} + \lambda\mathbf{B}$. Because

$M_{1,1} - M_{0,1} = \det(\mathbf{M}) \neq 0$ and $0 \leq M_{0,1} < M_{1,1} \leq 1$, we derive that $\lambda \in [0,1)$. Because $1 - M_{1,1} > 0$, we obtain that $0 < \frac{M_{0,1}}{1 - M_{1,1} + M_{0,1}} < 1$. Thus, $\mathbf{b} \in \Delta_\Sigma$. $\qquad \square$

## C.2 Proof of Claim 3

**Claim 3.** *In the binary case ($\Sigma = \{0(reject), 1(accept)\}$), the $\mathcal{M}^*$-invariant surprisal vector can be simplified as*

$$\mathrm{Surp}^*(\mathbf{w}, \mathbf{U}) = \frac{\mathbf{w} - \mathbf{q}}{\sqrt{q_0 q_1 (P_{1,1} - P_{0,1})}}.$$

*Proof of Claim 3.*

$$
\begin{aligned}
\det(\mathbf{U}) &= U_{0,0} U_{1,1} - U_{0,1} U_{1,0} \\
&= q_0 q_1 (P_{0,0} P_{1,1} - P_{0,1} P_{1,0}) \\
&= q_0 q_1 ((1 - P_{0,1}) P_{1,1} - P_{0,1}(1 - P_{1,1})) \\
&= q_0 q_1 (P_{1,1} - P_{0,1}).
\end{aligned}
$$

By plugging in the above formula into the definition of $\mathrm{Surp}^*$, we prove the claim. $\qquad \square$

## C.3 Proof of Theorem 1

**Theorem 1** (Invariance). *The $\mathcal{M}^*$-invariant surprisal is non-negative, vanishes when the state $\mathbf{w}$ equals the prior $\mathbf{q}$, and is invariant to $\mathbf{M}_{\lambda, \mathbf{b}}$ for all $\lambda \in [0,1)$, $\mathbf{b} \in \Delta_\Sigma$. The implication of invariance is that for any noise with $\lambda \in [0,1)$, $\mathrm{Surp}^*(\mathbf{w}, \mathbf{U}) = \mathrm{Surp}^*(\hat{\mathbf{w}}, \hat{\mathbf{U}})$.*

*Proof of Theorem 1.* We first show that $\mathbf{U}^{-1}$ is positive semi-definite, which implies the non-negativity. Afterwards, we prove the invariance.

**Claim 4.** $\mathbf{U}$ *and* $\mathbf{U}^{-1}$ *are positive semi-definite.*

*Proof of Claim 4.* As the number of papers is finite, we denote all $K$ states as $\mathbf{w}_1, \mathbf{w}_2, \cdots, \mathbf{w}_K$, where state $\mathbf{w}_i$ represents the probability distribution of a random reviewer's signal for paper $i$ under review. Let $1 \times K$ vector $\mathbf{q}_{\mathrm{state}}$ denote the prior distribution over the states. For all $s, t$, $U_{s,t} = \sum_k q_{\mathbf{w}_k} \mathbf{w}_k(s) \mathbf{w}_k(t)$. Let $\mathbf{W}$ be a $K \times |\Sigma|$ matrix that stacks the states, i.e., for all $k$, the $k^{th}$ row of $\mathbf{W}$ is $\mathbf{w}_k$. The above equation is equivalent to $\mathbf{U} = \mathbf{W}^\top \mathrm{diag}(\mathbf{q}_{\mathrm{state}}) \mathbf{W}$ thus $\mathbf{U}$ is positive semi-definite. Because a positive semi-definite matrix's inverse is also positive semi-definite. $\mathbf{U}^{-1}$ is also positive semi-definite. $\qquad \square$

**Claim 5.** $\hat{\mathbf{q}} = \mathbf{q}\mathbf{M}$, $\hat{\mathbf{w}} = \mathbf{w}\mathbf{M}$, $\hat{\mathbf{U}} = \mathbf{M}^\top \mathbf{U} \mathbf{M}$.

*Proof of Claim 5.* When noise $\mathbf{M}$ applies, $\forall t \in \Sigma, \hat{w}_t = \sum_s w_s M_{s,t}$, $\forall s, t \in \Sigma, \hat{U}_{s,t} = \sum_{a,b} U_{a,b} M_{a,s} M_{b,t}$. Thus, $\hat{\mathbf{w}} = \mathbf{w}\mathbf{M}$, $\hat{\mathbf{U}} = \mathbf{M}^\top \mathbf{U} \mathbf{M}$. The noisy prior is the marginal distribution of the noisy joint distribution. Let $\mathbf{1}^{1 \times |\Sigma|}$ denote a $|\Sigma|$-dimensional row vector with all elements equal to one. We have

$$
\begin{aligned}
\hat{\mathbf{q}} &= \mathbf{1}^{1 \times |\Sigma|} \hat{\mathbf{U}} \\
&= \mathbf{1}^{1 \times |\Sigma|} \mathbf{M}^\top \mathbf{U} \mathbf{M} && (\mathbf{M}^\top \text{ is column-stochastic}) \\
&= \mathbf{1}^{1 \times |\Sigma|} \mathbf{U} \mathbf{M} \\
&= \mathbf{q}\mathbf{M}
\end{aligned}
$$

$\qquad \square$

**Claim 6.** $\det(\mathbf{M}_{\lambda, \mathbf{b}}) = (1 - \lambda)^{|\Sigma| - 1}$

*Proof of Claim 6.*

$$\det(\mathbf{M}_{\lambda,\mathbf{b}}) = \det((1-\lambda)\mathbf{I} + \lambda\mathbf{B})$$

$$= (1-\lambda)^{|\Sigma|}\det(\mathbf{I} + \frac{\lambda}{1-\lambda}\mathbf{B}) \qquad (\det(c\mathbf{A}) = c^n\det(\mathbf{A}) \text{ where } \mathbf{A} \text{ is } n \times n \text{ matrix})$$

$$= (1-\lambda)^{|\Sigma|}\det(\mathbf{I} + \frac{\lambda}{1-\lambda}\mathbf{1}^\top\mathbf{b}) \qquad (\mathbf{1} = (1,1,\cdots,1))$$

$$= (1-\lambda)^{|\Sigma|}(1 + \frac{\lambda}{1-\lambda}\mathbf{b}\mathbf{1}^\top) \qquad \text{(Sylvester's determinant theorem)}$$

$$= (1-\lambda)^{|\Sigma|}(1 + \frac{\lambda}{1-\lambda}) \qquad (\mathbf{b} \text{ is a distribution vector thus all elements sum to 1.})$$

$$= (1-\lambda)^{|\Sigma|-1}$$

$\square$

Let $\hat{\mathbf{w}}, \hat{\mathbf{U}}$ denote the noisy state and joint distribution where the noise is $\mathbf{M} = \mathbf{M}_{\lambda,\mathbf{b}}$, we have

$$\mathrm{Surp}^*(\hat{\mathbf{w}}, \hat{\mathbf{U}}) = \det(\hat{\mathbf{U}})^{-\frac{1}{2(|\Sigma|-1)}}(\hat{\mathbf{w}} - \hat{\mathbf{q}})$$

$$= \det(\mathbf{M}^\top\mathbf{U}\mathbf{M})^{-\frac{1}{2(|\Sigma|-1)}}(\hat{\mathbf{w}} - \hat{\mathbf{q}})$$

$$= \det(\mathbf{U})^{-\frac{1}{2(|\Sigma|-1)}}\det(\mathbf{M})^{-\frac{1}{|\Sigma|-1}}(\hat{\mathbf{w}} - \hat{\mathbf{q}})$$

$$= \frac{1}{1-\lambda}\det(\mathbf{U})^{-\frac{1}{2(|\Sigma|-1)}}(\hat{\mathbf{w}} - \hat{\mathbf{q}}) \qquad \text{(Claim 6)}$$

$$= \frac{1}{1-\lambda}\det(\mathbf{U})^{-\frac{1}{2(|\Sigma|-1)}}(\mathbf{w} - \mathbf{q})\mathbf{M} \qquad \text{(Claim 5)}$$

$$= \det(\mathbf{U})^{-\frac{1}{2(|\Sigma|-1)}}(\mathbf{w} - \mathbf{q})\Big(\mathbf{I} + \frac{\lambda}{1-\lambda}\mathbf{B}\Big)$$

$$= \det(\mathbf{U})^{-\frac{1}{2(|\Sigma|-1)}}(\mathbf{w} - \mathbf{q}) + \frac{\lambda}{1-\lambda}\det(\mathbf{U})^{-\frac{1}{2(|\Sigma|-1)}}(\mathbf{w} - \mathbf{q})\mathbf{B}$$

$$= \det(\mathbf{U})^{-\frac{1}{2(|\Sigma|-1)}}(\mathbf{w} - \mathbf{q}) \qquad (\mathbf{w}\mathbf{B} = \mathbf{b} = \mathbf{q}\mathbf{B} \text{ by definition of } \mathbf{B})$$

$$= \mathrm{Surp}^*(\mathbf{w}, \mathbf{U}).$$

Thus, we finish the analysis for the invariant surprisal vector.

$\square$

## C.4 Proof of Theorem 2 and Theorem 3

In this section, we will show the omitted proof of Theorem 2 and Theorem 3. Before proving Theorem 2 and Theorem 3, we first analyse the results by assuming we know $\hat{\mathbf{P}}$ in all scenarios.

**Lemma 1** (Error probability, known $\hat{\mathbf{P}}$). *For any pair of papers $A, B$ whose clean states are $\mathbf{w}^A, \mathbf{w}^B$ correspondingly (without loss of generality, let $\mathbb{E}_{\mathbf{w}^A}\varphi < \mathbb{E}_{\mathbf{w}^B}\varphi$), given their noise matrix $\mathbf{M}^A, \mathbf{M}^B \in \mathcal{M}^*$ correspondingly,*

$$\Pr\left[\frac{\mathbb{E}_{\hat{\mathbf{v}}^A}\varphi - \mathbb{E}_{\hat{\mathbf{q}}^A}\varphi}{\det(\hat{\mathbf{U}}^A)^{\frac{1}{2(|\Sigma|-1)}}} \geq \frac{\mathbb{E}_{\hat{\mathbf{v}}^B}\varphi - \mathbb{E}_{\hat{\mathbf{q}}^B}\varphi}{\det(\hat{\mathbf{U}}^B)^{\frac{1}{2(|\Sigma|-1)}}} \Big| \mathbf{w}^A, \mathbf{w}^B\right] \leq \exp\left\{\frac{-2(\mathbb{E}_{\mathbf{w}^B}\varphi - \mathbb{E}_{\mathbf{w}^A}\varphi)^2\det(\mathbf{U})^{-\frac{1}{|\Sigma|-1}}}{(\varphi_{\max} - \varphi_{\min})^2\big(\frac{1}{n^A(1-\lambda^A)^2} + \frac{1}{n^B(1-\lambda^B)^2}\big)}\right\}$$

*Proof of Lemma 1.* Recall that $\hat{v}$ is the empirical frequency of distribution $\hat{w}$. When we have infinite number of reviewers, $\mathbb{E}_{\hat{\mathbf{v}}^A}\varphi = \mathbb{E}_{\hat{\mathbf{w}}^A}\varphi$ and $\mathbb{E}_{\hat{\mathbf{v}}^B}\varphi = \mathbb{E}_{\hat{\mathbf{w}}^B}\varphi$. Thus,

$$\Pr\left[\frac{\mathbb{E}_{\hat{\mathbf{v}}^A}\varphi-\mathbb{E}_{\hat{\mathbf{q}}^A}\varphi}{\det(\hat{\mathbf{U}}^A)^{\frac{1}{2(|\Sigma|-1)}}}\geq\frac{\mathbb{E}_{\hat{\mathbf{v}}^B}\varphi-\mathbb{E}_{\hat{\mathbf{q}}^B}\varphi}{\det(\hat{\mathbf{U}}^B)^{\frac{1}{2(|\Sigma|-1)}}}\Big|\mathbf{w}^A,\mathbf{w}^B\right]$$

$$=\Pr\left[\frac{\mathbb{E}_{\hat{\mathbf{w}}^A}\varphi-\mathbb{E}_{\hat{\mathbf{q}}^A}\varphi}{\det(\hat{\mathbf{U}}^A)^{\frac{1}{2(|\Sigma|-1)}}}\geq\frac{\mathbb{E}_{\hat{\mathbf{w}}^B}\varphi-\mathbb{E}_{\hat{\mathbf{q}}^B}\varphi}{\det(\hat{\mathbf{U}}^B)^{\frac{1}{2(|\Sigma|-1)}}}\Big|\mathbf{w}^A,\mathbf{w}^B\right]$$

$$=\Pr\left[\mathbb{E}_{\mathbf{w}^B}\varphi-\mathbb{E}_{\mathbf{w}^A}\varphi\geq0\right]\qquad\text{(Corollary 1: invariance)}$$

$$=1$$

However, when the number of reviewers is finite, we need to analyse the difference between $\hat{w}$ and $\hat{v}$ by concentration bound.

$$\Pr\left[\frac{\mathbb{E}_{\hat{\mathbf{v}}^A}\varphi-\mathbb{E}_{\hat{\mathbf{q}}^A}\varphi}{\det(\hat{\mathbf{U}}^A)^{\frac{1}{2(|\Sigma|-1)}}}\geq\frac{\mathbb{E}_{\hat{\mathbf{v}}^B}\varphi-\mathbb{E}_{\hat{\mathbf{q}}^B}\varphi}{\det(\hat{\mathbf{U}}^B)^{\frac{1}{2(|\Sigma|-1)}}}\Big|\mathbf{w}^A,\mathbf{w}^B\right]$$

$$=\Pr\left[\frac{\mathbb{E}_{\hat{\mathbf{v}}^A}\varphi}{\det(\hat{\mathbf{U}}^A)^{\frac{1}{2(|\Sigma|-1)}}}-\frac{\mathbb{E}_{\hat{\mathbf{v}}^B}\varphi}{\det(\hat{\mathbf{U}}^B)^{\frac{1}{2(|\Sigma|-1)}}}\geq\frac{\mathbb{E}_{\hat{\mathbf{q}}^A}\varphi}{\det(\hat{\mathbf{U}}^A)^{\frac{1}{2(|\Sigma|-1)}}}-\frac{\mathbb{E}_{\hat{\mathbf{q}}^B}\varphi}{\det(\hat{\mathbf{U}}^B)^{\frac{1}{2(|\Sigma|-1)}}}\Big|\mathbf{w}^A,\mathbf{w}^B\right]$$

$$=\Pr\left[\frac{\mathbb{E}_{\hat{\mathbf{v}}^A}\varphi-\mathbb{E}_{\hat{\mathbf{w}}^A}\varphi}{\det(\hat{\mathbf{U}}^A)^{\frac{1}{2(|\Sigma|-1)}}}-\frac{\mathbb{E}_{\hat{\mathbf{v}}^B}\varphi-\mathbb{E}_{\hat{\mathbf{w}}^B}\varphi}{\det(\hat{\mathbf{U}}^B)^{\frac{1}{2(|\Sigma|-1)}}}\geq-\frac{\mathbb{E}_{\hat{\mathbf{w}}^A}\varphi-\mathbb{E}_{\hat{\mathbf{q}}^A}\varphi}{\det(\hat{\mathbf{U}}^A)^{\frac{1}{2(|\Sigma|-1)}}}+\frac{\mathbb{E}_{\hat{\mathbf{w}}^B}\varphi-\mathbb{E}_{\hat{\mathbf{q}}^B}\varphi}{\det(\hat{\mathbf{U}}^B)^{\frac{1}{2(|\Sigma|-1)}}}\Big|\mathbf{w}^A,\mathbf{w}^B\right]$$

$$=\Pr\left[\frac{\mathbb{E}_{\hat{\mathbf{v}}^A}\varphi-\mathbb{E}_{\hat{\mathbf{w}}^A}\varphi}{\det(\hat{\mathbf{U}}^A)^{\frac{1}{2(|\Sigma|-1)}}}-\frac{\mathbb{E}_{\hat{\mathbf{v}}^B}\varphi-\mathbb{E}_{\hat{\mathbf{w}}^B}\varphi}{\det(\hat{\mathbf{U}}^B)^{\frac{1}{2(|\Sigma|-1)}}}\geq-\frac{\mathbb{E}_{\mathbf{w}^A}\varphi-\mathbb{E}_{\mathbf{q}}\varphi}{\det(\mathbf{U})^{\frac{1}{2(|\Sigma|-1)}}}+\frac{\mathbb{E}_{\mathbf{w}^B}\varphi-\mathbb{E}_{\mathbf{q}}\varphi}{\det(\mathbf{U})^{\frac{1}{2(|\Sigma|-1)}}}\Big|\mathbf{w}^A,\mathbf{w}^B\right]$$

$$\text{(Corollary 1: invariance of our score)}$$

$$=\Pr\left[\frac{\mathbb{E}_{\hat{\mathbf{v}}^A}\varphi-\mathbb{E}_{\hat{\mathbf{w}}^A}\varphi}{\det(\hat{\mathbf{U}}^A)^{\frac{1}{2(|\Sigma|-1)}}}-\frac{\mathbb{E}_{\hat{\mathbf{v}}^B}\varphi-\mathbb{E}_{\hat{\mathbf{w}}^B}\varphi}{\det(\hat{\mathbf{U}}^B)^{\frac{1}{2(|\Sigma|-1)}}}\geq\frac{\mathbb{E}_{\mathbf{w}^B}\varphi-\mathbb{E}_{\mathbf{w}^A}\varphi}{\det(\mathbf{U})^{\frac{1}{2(|\Sigma|-1)}}}\Big|\mathbf{w}^A,\mathbf{w}^B\right]$$

$$=\Pr\left[\frac{\mathbb{E}_{\hat{\mathbf{v}}^A}\varphi-\mathbb{E}_{\hat{\mathbf{w}}^A}\varphi}{\det(\mathbf{M}^A)^{\frac{1}{|\Sigma|-1}}}-\frac{\mathbb{E}_{\hat{\mathbf{v}}^B}\varphi-\mathbb{E}_{\hat{\mathbf{w}}^B}\varphi}{\det(\mathbf{M}^B)^{\frac{1}{|\Sigma|-1}}}\geq\mathbb{E}_{\mathbf{w}^B}\varphi-\mathbb{E}_{\mathbf{w}^A}\varphi\Big|\mathbf{w}^A,\mathbf{w}^B\right]\qquad\text{(Claim 5: }\hat{\mathbf{U}}=\mathbf{M}^\top\mathbf{U}\mathbf{M})$$

$$=\Pr\left[\frac{\mathbb{E}_{\hat{\mathbf{v}}^A}\varphi-\mathbb{E}_{\hat{\mathbf{w}}^A}\varphi}{1-\lambda^A}-\frac{\mathbb{E}_{\hat{\mathbf{v}}^B}\varphi-\mathbb{E}_{\hat{\mathbf{w}}^B}\varphi}{1-\lambda^B}\geq\mathbb{E}_{\mathbf{w}^B}\varphi-\mathbb{E}_{\mathbf{w}^A}\varphi\Big|\mathbf{w}^A,\mathbf{w}^B\right]$$

$$\text{(Claim 6: }\det(\mathbf{M}_{\lambda,\mathbf{b}})=(1-\lambda)^{|\Sigma|-1})$$

$$=\Pr\left[\left(\frac{\mathbb{E}_{\hat{\mathbf{v}}^A}\varphi}{1-\lambda^A}-\frac{\mathbb{E}_{\hat{\mathbf{v}}^B}\varphi}{1-\lambda^B}\right)-\mathbb{E}\left[\frac{\mathbb{E}_{\hat{\mathbf{v}}^A}\varphi}{1-\lambda^A}-\frac{\mathbb{E}_{\hat{\mathbf{v}}^B}\varphi}{1-\lambda^B}\right]\geq\mathbb{E}_{\mathbf{w}^B}\varphi-\mathbb{E}_{\mathbf{w}^A}\varphi\Big|\mathbf{w}^A,\mathbf{w}^B\right]$$

$$\text{(}\mathbb{E}_{\mathbf{w}}\varphi=\sum_s\varphi(s)w_s\text{ and }\hat{v}\text{ is the empirical frequency of distribution }\hat{w}\text{)}$$

We expand $\hat{v}^A$ and $\hat{v}^B$ as the sum of independent random variables.

$$\frac{\mathbb{E}_{\hat{\mathbf{v}}^A}\varphi}{1-\lambda^A}-\frac{\mathbb{E}_{\hat{\mathbf{v}}^B}\varphi}{1-\lambda^B}$$

$$=\frac{\sum_{i=1}^{n^A}\varphi(x_i^A)}{n^A(1-\lambda^A)}-\frac{\sum_{j=1}^{n^B}\varphi(x_j^B)}{n^B(1-\lambda^B)}$$

$$=\sum_{i=1}^{n^A}\frac{\varphi(x_i^A)}{n^A(1-\lambda^A)}-\sum_{j=1}^{n^B}\frac{\varphi(x_j^B)}{n^B(1-\lambda^B)}$$

Because each $x_i^A$ and $x_j^B$ is independent, we can define independent random variables $y_k (k \in [1, n^A + n^B])$ as

$$y_k = \begin{cases} \frac{\varphi(\hat{x}_k^A)}{n^A(1-\lambda^A)} & k \le n^A \\ \frac{-\varphi(\hat{x}_{k-n^A}^B)}{n^B(1-\lambda^B)} & k > n^A \end{cases}.$$

For $k \in [1, n^A]$, $\frac{\varphi_{\min}}{n^A(1-\lambda^A)} \le y_k \le \frac{\varphi_{\max}}{n^A(1-\lambda^A)}$. For $k \in [n^A+1, n^A+n^B]$, $-\frac{\varphi_{\max}}{n^B(1-\lambda^B)} \le y_k \le -\frac{\varphi_{\min}}{n^B(1-\lambda^B)}$. Using the variables $y_k (k \in [1, n^A + n^B])$, we can apply the standard Hoeffding's Inequality.

$$
\begin{aligned}
&\Pr\left[ \frac{\mathbb{E}_{\hat{\mathbf{v}}^A}\varphi - \mathbb{E}_{\hat{\mathbf{q}}^A}\varphi}{\det(\hat{\mathbf{U}}^A)^{\frac{1}{2(|\Sigma|-1)}}} \ge \frac{\mathbb{E}_{\hat{\mathbf{v}}^B}\varphi - \mathbb{E}_{\hat{\mathbf{q}}^B}\varphi}{\det(\hat{\mathbf{U}}^B)^{\frac{1}{2(|\Sigma|-1)}}} \Big| \mathbf{w}^A, \mathbf{w}^B \right] \\
&= \Pr\left[ \Sigma_{k=1}^{n^A+n^B} y_k - \mathbb{E}\left[ \Sigma_{k=1}^{n^A+n^B} y_k \right] \ge \mathbb{E}_{\mathbf{w}^B}\varphi - \mathbb{E}_{\mathbf{w}^A}\varphi | \mathbf{w}^A, \mathbf{w}^B \right] \\
&\le \exp\left\{ -\frac{2(\mathbb{E}_{\mathbf{w}^B}\varphi - \mathbb{E}_{\mathbf{w}^A}\varphi)^2}{n^A\left(\frac{\varphi_{\max}-\varphi_{\min}}{n^A(1-\lambda^A)}\right)^2 + n^B\left(\frac{\varphi_{\max}-\varphi_{\min}}{n^B(1-\lambda^B)}\right)^2} \right\} \qquad \text{(Hoeffding's Inequality)} \\
&= \exp\left\{ -\frac{2(\mathbb{E}_{\mathbf{w}^B}\varphi - \mathbb{E}_{\mathbf{w}^A}\varphi)^2}{(\varphi_{\max}-\varphi_{\min})^2\left(\frac{1}{n^A(1-\lambda^A)^2} + \frac{1}{n^B(1-\lambda^B)^2}\right)} \right\}
\end{aligned}
$$

$\square$

**Theorem 2** (Error probability, binary case). *For any pair of papers $A, B$ whose clean states are $\mathbf{w}^A, \mathbf{w}^B$ correspondingly (without loss of generality, let $w_1^A < w_1^B$), given their noise matrices $\mathbf{M}^A, \mathbf{M}^B \in \mathcal{M}^*$ correspondingly, the error probability $\Pr[S(A) > S(B)|w_1^A, w_1^B] + \frac{1}{2}\Pr[S(A) = S(B)|w_1^A, w_1^B]$ goes to 0 when the number of reviewers $n_A, n_B$ goes to infinity, and is bounded by*

$$\left(\hat{w}_1^A\right)^{n_A} + \left(1-\hat{w}_1^B\right)^{n_B} - \frac{1}{2}\left[\left(\hat{w}_1^A\right)^{n_A}\left(\hat{w}_1^B\right)^{n_B} + \left(1-\hat{w}_1^A\right)^{n_A}\left(1-\hat{w}_1^B\right)^{n_B}\right] + \exp\left\{ -\frac{2(w_1^B-w_1^A)^2}{\frac{1}{n_A(1-\lambda_A)^2} + \frac{1}{n_B(1-\lambda_B)^2}} \right\},$$

*where $\lambda_A$ is the noise level of paper A and $\lambda_B$ is the noise level of paper B[14].*

*Proof of Theorem 2.* We will show that with high probability, the prediction matrix can be constructed. Once it is constructed, we use the result of Lemma 1.

---

[14]Note that the bound is independent of the biases.

$$\frac{1}{2}\Pr[S(A)=S(B)|w_1^A,w_1^B]+\Pr[S(A)>S(B)|w_1^A,w_1^B]$$

$$=\frac{1}{2}\Pr[S(A)=+\infty\cap S(B)=+\infty|w_1^A,w_1^B]+\frac{1}{2}\Pr[S(A)=-\infty\cap S(B)=-\infty|w_1^A,w_1^B]$$

$$+\frac{1}{2}\Pr[S(A)=S(B)\cap S(A)\neq\infty\cap S(B)\neq\infty|w_1^A,w_1^B]+\Pr[S(A)=+\infty\cap S(B)\neq+\infty|w_1^A,w_1^B]$$

$$+\Pr[S(A)\neq-\infty\cap S(B)=-\infty|w_1^A,w_1^B]+\Pr[S(A)>S(B)\cap S(A)\neq\infty\cap S(B)\neq\infty|w_1^A,w_1^B]$$

$$\leq\frac{1}{2}\Pr[S(A)=+\infty\cap S(B)=+\infty|w_1^A,w_1^B]+\frac{1}{2}\Pr[S(A)=-\infty\cap S(B)=-\infty|w_1^A,w_1^B]$$

$$+\Pr[S(A)=+\infty\cap S(B)\neq+\infty|w_1^A,w_1^B]+\Pr[S(A)\neq-\infty\cap S(B)=-\infty|w_1^A,w_1^B]$$

$$+\Pr[S(A)\geq S(B)\cap S(A)\neq\infty\cap S(B)\neq\infty|w_1^A,w_1^B]$$

$$=\frac{1}{2}\left(\hat{w}_1^A\right)^{n^A}\left(\hat{w}_1^B\right)^{n^B}+\frac{1}{2}\left(1-\hat{w}_1^A\right)^{n^A}\left(1-\hat{w}_1^B\right)^{n^B}+\left(\hat{w}_1^A\right)^{n^A}\left(1-\left(\hat{w}_1^B\right)^{n^B}\right)$$

$$+\left(1-\left(1-\hat{w}_1^A\right)^{n^A}\right)\left(1-\hat{w}_1^B\right)^{n^B}+\Pr[S(A)\geq S(B)\cap S(A)\neq\infty\cap S(B)\neq\infty|w_1^A,w_1^B]]$$

$$=\left(\hat{w}_1^A\right)^{n^A}+\left(1-\hat{w}_1^B\right)^{n^B}-\frac{1}{2}\left[\left(\hat{w}_1^A\right)^{n^A}\left(\hat{w}_1^B\right)^{n^B}+\left(1-\hat{w}_1^A\right)^{n^A}\left(1-\hat{w}_1^B\right)^{n^B}\right]$$

$$+\Pr[S(A)\geq S(B)\cap S(A)\neq\infty\cap S(B)\neq\infty|w_1^A,w_1^B]]$$

$$\leq\left(\hat{w}_1^A\right)^{n^A}+\left(1-\hat{w}_1^B\right)^{n^B}-\frac{1}{2}\left[\left(\hat{w}_1^A\right)^{n^A}\left(\hat{w}_1^B\right)^{n^B}+\left(1-\hat{w}_1^A\right)^{n^A}\left(1-\hat{w}_1^B\right)^{n^B}\right]$$

$$+\Pr\left[\frac{\hat{v}^A-\hat{q}^A}{\sqrt{\det(\hat{\mathbf{U}}^A)}}\geq\frac{\hat{v}^B-\hat{q}^B}{\sqrt{\det(\hat{\mathbf{U}}^B)}}\Big|w_1^A,w_1^B\right]$$

$$\leq\left(\hat{w}_1^A\right)^{n^A}+\left(1-\hat{w}_1^B\right)^{n^B}-\frac{1}{2}\left[\left(\hat{w}_1^A\right)^{n^A}\left(\hat{w}_1^B\right)^{n^B}+\left(1-\hat{w}_1^A\right)^{n^A}\left(1-\hat{w}_1^B\right)^{n^B}\right]$$

$$+\exp\left\{-\frac{2(w_1^B-w_1^A)^2}{\left(\frac{1}{n^A(1-\lambda^A)^2}+\frac{1}{n^B(1-\lambda^B)^2}\right)}\right\} \qquad\text{(Lemma 1)}$$

$$\square$$

**Theorem 3** (Error probability, general case). *For any pair of tasks $A,B$ whose clean states are $\mathbf{w}^A,\mathbf{w}^B$ correspondingly (without loss of generality, let $\mathbb{E}_{\mathbf{w}^A}\varphi<\mathbb{E}_{\mathbf{w}^B}\varphi$), given their noise matrices $\mathbf{M}^A,\mathbf{M}^B\in\mathcal{M}^*$ correspondingly, the error probability $\Pr[S(A)>S(B)|\mathbf{w}^A,\mathbf{w}^B]+\frac{1}{2}\Pr[S(A)=S(B)|\mathbf{w}^A,\mathbf{w}^B]$ goes to $0$ when the number of reviewers $n_A,n_B$ goes to infinity, and is bounded by*[15]

$$\sum_{s\in\Sigma}(1-\hat{w}_s^A)^{n_A}+\sum_{s\in\Sigma}(1-\hat{w}_s^B)^{n_B}+\exp\left\{-\frac{2(\mathbb{E}_{\mathbf{w}^B}\varphi-\mathbb{E}_{\mathbf{w}^A}\varphi)^2}{(\varphi_{\max}-\varphi_{\min})^2\left(\frac{1}{n_A(1-\lambda_A)^2}+\frac{1}{n_B(1-\lambda_B)^2}\right)}\right\}.$$

*Proof of Theorem 3.* First, we give an upper bound of the probability that the score is undefined.

$$\Pr[S(A)\text{ is undefined}]$$

$$=\Pr[\exists s\in\Sigma,\forall i\in[n^A],\hat{x}_i\neq s]$$

$$\leq\sum_{s\in\Sigma}\Pr[\forall i\in[n^A],\hat{x}_i\neq s]$$

$$=\sum_{s\in\Sigma}(1-\hat{w}_s^A)^{n^A}$$

Similarly, $\Pr[S(B)\text{ is undefined}]\leq\sum_{s\in\Sigma}(1-\hat{w}_s^B)^{n^B}$. We have proved that with high probability, the prediction matrix can be constructed thus the scores are not undefined. We then use the result

---

[15] $\varphi_{\max}$ and $\varphi_{\min}$ are the abbreviations of $\max_{s\in\Sigma}\varphi(s)$ and $\min_{s\in\Sigma}\varphi(s)$ respectively.

of Lemma 1.

$$\Pr[S(A) > S(B) | w_1^A, w_1^B] + \frac{1}{2} \Pr[S(A) = S(B) | w_1^A, w_1^B]$$

$$\leq \Pr[S(A) \text{ is undefined}] + \Pr[S(B) \text{ is undefined}]$$

$$+ \Pr[S(A) \geq S(B) \cap \text{both } S(A) \text{ and } S(B) \text{ are not undefined} | w_1^A, w_1^B]$$

$$\leq \sum_{s \in \Sigma} (1 - \hat{w}_s^A)^{n^A} + \sum_{s \in \Sigma} (1 - \hat{w}_s^B)^{n^B}$$

$$+ \Pr\left[ \frac{\mathbb{E}_{\hat{\mathbf{v}}^A} \varphi - \mathbb{E}_{\hat{\mathbf{q}}^A} \varphi}{\det(\hat{\mathbf{U}}^A)^{\frac{1}{2(|\Sigma|-1)}}} \geq \frac{\mathbb{E}_{\hat{\mathbf{v}}^B} \varphi - \mathbb{E}_{\hat{\mathbf{q}}^B} \varphi}{\det(\hat{\mathbf{U}}^B)^{\frac{1}{2(|\Sigma|-1)}}} \,\middle|\, \mathbf{w}^A, \mathbf{w}^B \right]$$

$$\leq \sum_{s \in \Sigma} (1 - \hat{w}_s^A)^{n^A} + \sum_{s \in \Sigma} (1 - \hat{w}_s^B)^{n^B}$$

$$+ \exp\left\{ -\frac{2(\mathbb{E}_{\mathbf{w}^B} \varphi - \mathbb{E}_{\mathbf{w}^A} \varphi)^2}{(\varphi_{\max} - \varphi_{\min})^2 \left( \frac{1}{n^A(1-\lambda^A)^2} + \frac{1}{n^B(1-\lambda^B)^2} \right)} \right\} \qquad \text{(Lemma 1)}$$

$$\square$$

# D  SP-inspired Score

In this section, we introduce a new benchmark for comparison, inspired by the Surprising Popular (SP) mechanism, which we term as the SP-inspired score. The Surprising Popular mechanism is designed to consolidate multiple reports into a single assertion of which world state we are in, rather than generating scores to compare different alternatives. Utilizing the notations explained earlier in this paper, mapping $\varphi : \Sigma \to \mathbb{R}$ represents each signal's quality, $\hat{v}_s = \frac{1}{n} \sum_{i=1}^n \mathbf{1}[\hat{x}_i = s]$ stands for the frequency of the noisy signal $s$, and $\hat{q}_s = (\sum_t \frac{\hat{P}_{s,t}}{\hat{P}_{t,s}})^{-1}$ represents the reconstructed prior probability of the noisy signal $s$. When dealing with binary signals ($\Sigma = \{0,1\}$), the SP mechanism presumes that the true state is $1$ if $\frac{\hat{v}_1}{\hat{q}_1} > \frac{\hat{v}_0}{\hat{q}_0}$; otherwise, it is presumed that the true state is $0$. In general settings, the SP mechanism determines the true state as the signal $s$ that maximizes $\frac{\hat{v}_s}{\hat{q}_s}$.

To facilitate comparisons between alternatives, we define the SP-inspired score as

$$S((\hat{x}_i, \hat{\mathbf{p}}_i)_{i \in [n]}) = \sum_{s \in \Sigma} \varphi(s) \frac{\hat{v}_s}{\hat{q}_s}.$$

In the case of binary signals, we set $\varphi(0) = -1$ and $\varphi(1) = 1$. This results in the SP-inspired score being $S((\hat{x}_i, \hat{\mathbf{p}}i)i \in [n]) = \frac{\hat{v}_1}{\hat{q}_1} - \frac{\hat{v}_0}{\hat{q}_0}$.

**Difference between Surprisal-based Score and SP-inspired Score**  Recall our Empirical Surprisal-based Score is defined as

$$S((\hat{x}_i, \hat{\mathbf{p}}_i)_{i \in [n]}) = (\mathbb{E}_{\hat{\mathbf{v}}} \varphi - \mathbb{E}_{\hat{\mathbf{q}}} \varphi) \det(\hat{\mathbf{U}})^{-\frac{1}{2(|\Sigma|-1)}}.$$

Both our Surprisal-based Score and the SP-inspired score involve reconstructing the prior from agents' predictions. However, they differ in how to exploit this reconstructed prior to obtain a calibrated score. Our Surprisal-based Score is invariant of noise, meaning it does not change in the presence of noise when the number of reviewers is infinite. On the other hand, the SP-inspired score is affected by noise. This implies that in certain situations, even when we have an infinite number of reviewers, the SP-inspired score fails to distinguish which of the two papers is better. However, our Surprisal-based Score exhibits lower stability compared to the SP-inspired score, because it involves a division of the correlation, which may lead to a high variance when the number of reviewers is small.

**Numerical Experiments**  We perform numerical experiments to compare the performance of our Surprisal-based Score and the SP-inspired score in the binary setting ($\Sigma = \{0(\text{reject}), 1(\text{accept})\}$). We employ the same experimental setup as described in Section 6. Figure 6 and Figure 7 present the experimental results. Upon analyzing the results, we observed that the performance of our Surprisal-based Score and the SP-inspired Score is roughly the same when the number of reviewers is small.

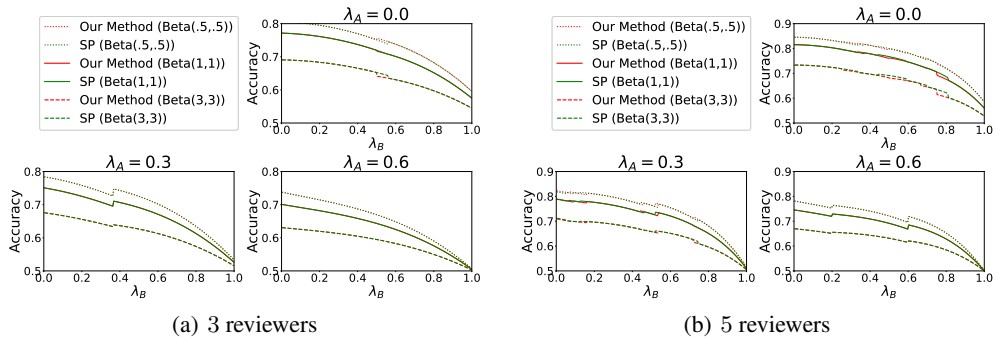

Figure 6: **Performance evaluation (opposite biases)**

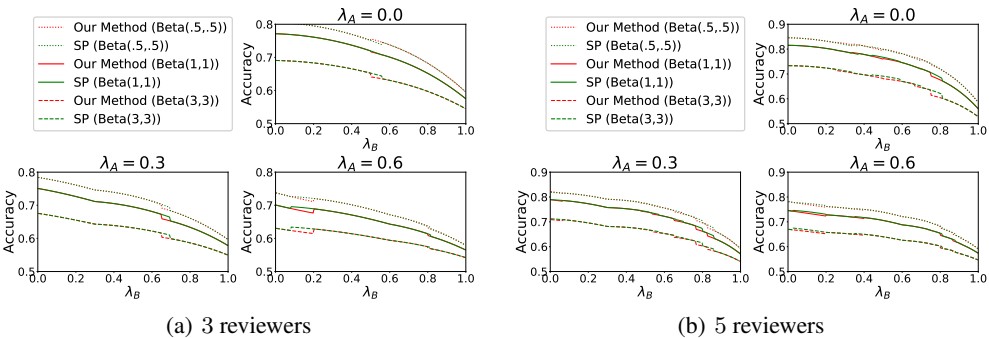

Figure 7: **Performance evaluation (same bias)**

