# OpenReview forum: "Calibrating “Cheap Signals” in Peer Review without a Prior"
_NeurIPS.cc/2023/Conference — NeurIPS 2023 poster_

### Official Review · Reviewer_VrqZ · 2023-07-04

**Soundness:** 3 good
**Presentation:** 2 fair
**Contribution:** 2 fair
**Rating:** 5
**Confidence:** 2

**Summary:**

The paper tackles a problem in peer review where reviewers may provide noisy/biased ratings for papers. The paper investigate a "one-shot" scoring process inspired by the "Surprisingly Popular" method, that can rank papers by their true quality without any prior knowledge, even if different papers have different noise levels and biases.

Their method relies on eliciting reviewers’ predictions for a random reviewer’s signal and using this for calibration. Specifically, their model assumes that there is a prior distribution over the states (where each paper's state in the "clean" setting is the distribution of reviewers’ signals)  and that reviewers are perfect Bayesians who can update their beliefs based on their private signals. In the "noisy" setting, reviewers can only observe noisy versions of their signals, determined by a random mapping that depends on the noise level and the bias vector.

Then, the paper defines the true quality of a paper as the expected score in the clean setting, and aims to design a method that can rank papers by their true quality with high probability, even if they have different noises/biases. For this, they introduce a surprisal vector that measures how "surprising" a paper’s state is compared to the prior distribution, and is normalized by the correlation between reviewers’ signals. The key result is that the surprisal vector is noise-invariant, i.e., the surprisal vector is the same in the clean and the noisy setting, regardless of the noise model.  Then, by computing an empirical surprise-based score for each paper, the mechanism papers can be compared in a noise-invariant way and are consistent with their true qualities.

**Strengths:**

- The paper is tackling an important problem, which is the question of how to de-bias and de-noise reviewers to obtain a clearer view of paper quality.

- The observation that the surprisal vector is invariant under noise is insightful and interesting, as was the idea to look to the surprisingly popular mechanism.

**Weaknesses:**

- The presentation of the paper could be improved. The intro feels a bit unfocused, there are a number of grammatical errors, and the paper is a bit notation heavy. It might help to move the notation table from the appendix to the main body, and also expand on the "high level ideas" to more provide a clearer overview of the proposed method. It took a couple reads to understand exactly what was being proposed.

- The assumption of calibrated agent predictions is not clearly justifiable in a context where agent signals are assumed to be potentially biased.

- In line 162, the result cited by Kong et al. (2018) is outdated, suggest updating with more recent results that improve on this:

Schoenebeck, Grant, and Fang-Yi Yu. "Two strongly truthful mechanisms for three heterogeneous agents answering one question." ACM Transactions on Economics and Computation 10.4 (2023): 1-26.

Srinivasan, Siddarth, and Jamie Morgenstern. "Auctions and prediction markets for scientific peer review." arXiv preprint arXiv:2109.00923 (2021).



**Questions:**

- How do you handle det(U) being negative in practice?

- Can you give some intuition for why is the normalization of the surprisal vector important?

- Do the exponential bounds in the general case not depend on the cardinality of the signal space? Can you give some intuition for why this is the case?

- Are the results robust if noise in agents' predictions of other agents' signals correlated with the noise in their own signals?

**Limitations:**

The main limitation I see is that it is unclear what to do if some of the computed statistics (like det(U)) violate the non-negativity requirement.

The authors are up front about other limitations. These aren't ideal, but may be OK for the goals of this paper.
- Authors assume no incentive issues, i.e., that all agents report truthfully. However, this is unlikely to be the case in practice, especially if participants know that there is

- The authors assume that agents are calibrated in their predictions, even if their signals may be biased. This does feel like a bit of an odd assumption, and is unlikely to be true. Why would signals be biased, but predictions of others' signals be calibrated?

---

> ### Author Rebuttal · Authors · 2023-08-10
>
> ## Reviewer VrqZ
>
> Thank you for your insightful comments and suggestions.
>
> ### W1: The presentation of the paper could be improved
>
> In the final version, we will relocate the symbol clarifications (Table 1, currently in the appendix on page 17) back to section 2 for ease of reference. Besides, we will provide a more detailed and clear overview in "high level ideas". Thank you for valuable suggestions.
>
>
> ### W3: Outdated result cited by Kong et al. (2018)
>
> Thanks for pointing out, we will update these references in the final version. In addition, we will make the related work section more detailed in the final version.
>
>
> ### Q1: How do you handle det(U) being negative in practice
>
> In our model, when reviewers are Bayesian, det(U) will never become negative. In practice, a negative det(U) suggests that when a reviewer favors a paper, her belief in the likelihood of another reviewer also favoring it decreases. This unusual scenario signals to the Program Committee that this particular paper warrants further discussion and careful consideration before making a final decision.
>
>
> ### Q2: Can you give some intuition for why is the normalization of the surprisal vector important
>
> Recall that in the binary case, our score is defined as (baseline score - prior expected scores) / correlation.
>
> First, the normalization constant adaptively compensates for varying noise levels. When faced with noisy reviewers providing weak signals beyond the prior, the baseline and prior expected scores will closely align, resulting in a correlation near zero. By normalizing using this small correlation, the overall scores are scaled up to compensate for the strong noise and weak signals. Conversely, when noise is weak and the reviewers' signals are strong, there is a clear distinction between the baseline and prior expected scores, leading to a higher correlation. This adaptive normalization ensures the metric's efficacy across contexts with varying noise levels.
>
> Second, only with the normalization, we can have the invariance results. The core idea is that a metric can only enable quantitative comparisons across different papers if and only if it is invariant to systematically biased noise. Without invariance, biased noise can easily distort comparisons for certain papers. For instance, with one noise pattern, paper A outscore paper B, while with a different biased noise pattern, paper B outscore paper A.
>
>
> ### Q3: Do the exponential bounds in the general case not depend on the cardinality of the signal space? Can you give some intuition for why this is the case
>
> In general cases, there exists a "natural" mapping $\varphi$ that allocates a real-valued score to each potential signal. For example, in NeurIPS, this mapping is represented as $r \rightarrow 3, br \rightarrow 4, ba \rightarrow 5, wa \rightarrow 6, a \rightarrow 7$. Such a "natural" mapping is prevalent in most peer review conferences. By employing this mapping, we can apply the concentration bound on the average mapping outcome, which is expressed solely in terms of the value range $\varphi_{\text{max}}-\varphi_{\text{min}}$.
>
>
> ### Q4: Are the results robust if noise in agents' predictions of other agents' signals correlated with the noise in their own signals
>
> Our assumption in the paper is that the reviewers' biases towards a paper consistently reflect in their predictions about others. If your question is, "Is the result robust if this correlation is not perfect Bayesian?", then the response would be that our score loses its theoretical guarantee in such cases. Nonetheless, our scores of the papers still provide meaningful insights for the Program Committee, particularly when assessing papers of borderline quality.
>
>
> ### L1: Authors assume no incentive issues
>
> The research question this paper attempts to address is "how to aggregate the biased evaluations in peer review, assuming evaluations are truthful". This research question is parallel and complementary to the research question "how to design incentive mechanisms to obtain truthful evaluations". We can employ mechanisms aimed at eliciting truthful evaluations (for example, the papers you mentioned) to ensure truthfulness, and then use our proposed method for aggregation. We will clarify this in the discussion section in the final version.
>
>
> ### L2: Why would signals be biased, but predictions of others' signals be calibrated
>
> In our paper, we interpret the scenario where a reviewer provides biased evaluation as she may not exert sufficient effort and only have access to "cheap signals". However, she can still report calibrated predictions concerning other reviewers’ evaluation, based on the signal she has. This scenario mirrors the Surprisingly Popular (SP) method [1]. In SP, it is assumed that while agents might give a biased response to a question like "what is the capital of Illinois", they can still provide a Bayesian prediction regarding other people’s answers based on their own response.
>
> [1] Prelec, Dražen, H. Sebastian Seung, and John McCoy. "A solution to the single-question crowd wisdom problem." *Nature* 541.7638 (2017): 532-535.

---

> > ### Comment · Reviewer_VrqZ · 2023-08-19
> >
> > I thank the author for their clarifications, and I have no further questions.

---

### Official Review · Reviewer_tv7v · 2023-07-07

**Soundness:** 4 excellent
**Presentation:** 4 excellent
**Contribution:** 3 good
**Rating:** 7
**Confidence:** 2

**Summary:**

This paper proposes a method to calculate peer-reviews scores for papers in the presence of systematically biased noise, such that the score of a paper with a higher expected score in a noise-free regime is higher than the score of a paper with a lower expected score in a noise-free regime with high probability (as the number of peer reviewers grows large). The method does not use reviewers' historical scores to compute their priors. Instead, the method asks reviewers to predict the scores of other reviewers (similar to the Bayesian truth serum). Numerical experiments confirm the theoretical results that the proposed score is better able to distinguish between "good" and "bad" papers under systematically biased noise compared to a baseline that just averages the uncalibrated peer review scores.

**Post-rebuttal:** I am raising my presentation score to 4 (excellent).

**Strengths:**

The paper is clear and precise, and spends sufficient time setting up the problem and building intuition. The proposed method adapts a well-known idea (Bayesian truth serum) to the setting of systematically-biased noise in peer review in a principled manner; this is novel to the best of my knowledge. It is also non-trivial, and provides a useful way to construct peer-review scores without needing historical data for each peer reviewer.

**Weaknesses:**

The generality of the exposition also makes it a bit difficult to follow. I think just limiting the main body of the paper to the binary decision case (accept, reject) and adjusting the notation for this (eg. a for accept, r for reject) would make it easier to follow the main argument of the paper. The generalization to multi-valued decisions (eg. review scores) could be relegated to an appendix or to just one theorem.

The numerical experiments do not consider SP as a baseline (possible with some modifications since it cannot be directly applied). A naive application of SP would be the most related method in prior work for the problem considered in this paper. Hence, it would help to see how this would perform.

**Questions:**

Q. Could the authors report numerical results from a naive application of SP as a baseline?

**Limitations:**

The authors have noted the limitations of their work.

---

> ### Author Rebuttal · Authors · 2023-08-10
>
> ## Reviewer tv7v
>
> Thank you for your insightful comments and suggestions.
>
> ### W1: Suggestion for limiting the main body of the paper to the binary decision case:
>
> Thank you for your suggestion. We will adopt it in the final version. This allows us to focus the main body on the binary signal case, making it easier for readers to follow the paper's flow and understand the core concepts.
>
> ### Q1: Could the authors report numerical results from a naive application of SP as a baseline
>
> The Surprisingly Popular (SP) method can be applied in our setting [1]. The SP method calculates the "prediction-normalized vote" for all signals $i$ as $\frac{w_i}{q_i}$ and asserts that the signal $\arg\max_i \frac{w_i}{q_i}$ is the best signal. Thus, when signals are binary ('reject': 0, 'accept': 1), the SP score of a paper is given by the "prediction-normalized vote" of 'accept' minus that of 'reject', i.e., $\frac{w_1}{q_1}-\frac{w_0}{q_0}$. However, the SP score is not invariant to noise. In some instances, even with infinite reviewers, the SP score fails to determine the relative quality of two papers (a specific example is provided below).
>
> [1] Prelec, Dražen, H. Sebastian Seung, and John McCoy. "A solution to the single-question crowd wisdom problem." *Nature* 541.7638 (2017): 532-535.
>
> **An example that the SP score fails to work**
>
> * Signals are binary ('reject'/'accept').
> * The prior over the states is $w_1\sim \text{Uniform}(0,1)$ (paper’s quality is distributed uniformly).
> * Paper A has true state $\mathbf{w}_A=[1/3,2/3]$, i.e., paper A has true quality $2/3$.
> * Paper A receives no noise, i.e., the noise matrix of paper A is $\mathbf{M}_A=\left[\begin{matrix}1 & 0\\\\0 & 1\\\\\\end{matrix}\right]$.
> * Paper B has true state $\mathbf{w}_B=[1/4,3/4]$, i.e., paper B has true quality $3/4$, which is better than paper A.
> * Paper B receives positive noise because its writing is good. The noise matrix of paper B is $\mathbf{M}_B=\left[\begin{matrix}1/3 & 2/3\\\\0 & 1\\\\\\end{matrix}\right]$.
>
> For paper A, we can calculate that $\hat{\mathbf{w}}_A=[1/3,2/3]\, \hat{\mathbf{q}}_A=[1/2,1/2]$. Thus the SP score of paper A is $\frac{2/3}{1/2}-\frac{1/3}{1/2}=\frac{2}{3}$.
>
> For paper B, we can calculate that $\hat{\mathbf{w}}_B=[1/12,11/12]\, \hat{\mathbf{q}}_B=[1/6,5/6]$. Thus the SP score of paper B is $\frac{11/12}{5/6}-\frac{1/12}{1/6}=\frac{3}{5}$.
>
> As a result, paper B has a better true quality than paper A, but has a lower SP score, even if there are an infinite number of reviewers.

---

> > ### Comment · Reviewer_tv7v · 2023-08-12
> >
> > Thanks! This is very helpful. Why not add SP to Figures 4 and 5?

---

> > > ### Author Response · Authors · 2023-08-14
> > >
> > > ### Using SP for comparison
> > > Thank you for your feedback!
> > > The method proposed by Prelec et al. (2017) was designed for aggregating multiple reports into a single decision, rather than generating scores for comparing two alternatives. Specifically, when signals are binary, their approach outputs 1 if $\frac{w_1}{q_1}>\frac{w_0}{q_0}$, and 0 otherwise. While effective for combining evaluations into a single decision, this does not directly produce scores that can be used to compare the relative quality of two papers or alternatives.
> > >
> > > To enable such comparisons, we propose a new score based on the surprisingly popular idea: $\frac{w_1}{q_1}-\frac{w_0}{q_0}$: this measures the amount of surprisal for 'accept', subtracting the amount of surprisal for 'reject'. We call it the SP-inspired score.
> > > We run the same numerical experiments as Figure 4 and 5. Since there is no place to attach figures, we describe the simulation results in words.
> > > * SP-inspired score and the surprisal based score have similar performance in the settings considered by Figures 4 and 5 when the number of reviewers is small like $n=3,5$, and the surprisal based score outperforms the SP-inspired score as the number of reviewers increase. Both of them are better than the simple average.
> > >
> > > Conceptually, the example we provided before demonstrates that the SP-inspired score does not calibrate based on the amount of noise. Without calibrating for noise amount, the SP-inspired score will introduce substantial bias against papers with high-quality but more noisy evaluations. Because of the lack of noise calibration, the error rate of the SP-inspired score does not converge to zero as the number of reviewers increases. This makes it challenging to provide theoretical guarantees on the SP-inspired score's performance across all cases. In contrast, the surprisal-based score does calibrate, allowing its error rate to decrease with more reviewers, backed by theoretical guarantees.
> > >
> > > However, without calibration, while suffering from bias, the SP-inspired score has a lower variance. When reviewer numbers are small, the SP-inspired score achieves similar average performance to the surprisal-based score.
> > >
> > > In the final version, we will include the SP-inspired score and expanded comparison results to further illustrate the strengths and limitations of both methods.

---

> > > > ### Comment · Reviewer_tv7v · 2023-08-14
> > > >
> > > > This is illuminating, it would be helpful to have the SP score in the appendix or main body depending on space constraints, I have raised my presentation score by 1 point. Thank you for your responses.

---

### Official Review · Reviewer_6t8A · 2023-07-08

**Soundness:** 3 good
**Presentation:** 1 poor
**Contribution:** 2 fair
**Rating:** 5
**Confidence:** 2

**Summary:**

The paper considers the problem of comparing two papers based on noisy ratings, where the noise can be arbitrarily biased for different papers. The paper elicits from each reviewer both a rating and a distribution of predicted ratings from other reviewers’ (based on a Bayesian update of the common prior, known to the reviewers). The paper proposes a scoring method that recovers the correct comparison between papers from the noisy reports, which essentially corrects the reported ratings using the reported prediction distributions. The authors theoretically show a bound on the error probability of a comparison based on their scoring method, and experimentally evaluate the comparison accuracy.

**Strengths:**

- The problem considered by the paper, calibrating biased noise in a one-shot comparison between items, is conceptually interesting. The authors motivate this problem well in the peer review setting with examples of cases where multiple reviewers may have the same bias.
- The authors prove a theoretical bound on the convergence rate of their method’s error probability to 0. The experimental settings considered are thorough (and demonstrate aspects of the theoretical result).


**Weaknesses:**

- While the problem itself is interesting, the assumption that all agents share and know a common noisy prior is very strong; e.g., in the peer review examples in the introduction, while the noise may be biased in the same direction for all reviewers, the noisy prior may not even be similar for all reviewers. Given this assumption, I’m unsure about the significance of the proposed method, which essentially recovers this prior (as in past work) and uses it to standardize the reports.
- I found the clarity of the writing to be generally poor, leading to some significant confusion at points (particularly in Sections 1 and 2). For example, in Section 1, I did not understand the claim following Example 2 (that a higher noise level would result in a lower expected score).  In Section 2, it was difficult to follow which aspects are observable by agents and by the mechanism.


**Questions:**

- Could the authors expand on what makes the setting of this work unique as compared to the settings of the past works referenced in Section 1.1 (Lines 123-129)? Are none of these works applicable as a baseline for comparison?
- The invariant surprisal vector concept is given a lot of focus (Section 3), when the purpose of this concept seems to only be that it’s used to prove Corollary 1. Since the surprisal-based score (in Section 4) is the main contribution, I might suggest making it more clear in Section 3 what the purpose of the invariant surprisal vector is, in order to keep the focus on the main contributions.
- Could the authors clarify the aspects of Section 1 and 2 referenced above?


**Limitations:**

The authors adequately address the limitations of the assumptions made (perfect Bayesian and identical reviewers).

---

> ### Author Rebuttal · Authors · 2023-08-10
>
> ## Reviewer 6t8A
>
> Thank you for your insightful comments and suggestions.
>
> ### W1: Assumption of common noisy prior is very strong
>
> We interpret the scenario where different reviewers have different priors as if they share a common prior but possess distinct private information. The "common prior assumption" is a foundational hypothesis in economics [1]. Works that employ this assumption model agents' differences as "information asymmetry" rather than "prior asymmetry". In other words, they assume that differences among agents arise from variations in their private information, rather than differences in their prior beliefs [2].
>
> [1] Morris, Stephen. "The common prior assumption in economic theory." *Economics & Philosophy* 11.2 (1995): 227-253.
>
> [2] Aumann, Robert J. "Agreeing to Disagree." *The Annals of Statistics* (1976): 1236-1239.
>
> ### Q1: What makes the setting of this work unique as compared to the settings of the past works
>
> In Section 1.1, we classify related work into two primary categories: "bias in peer review" and "reducing bias via second-order information".
>
> Regarding the "bias in peer review" category, our objectives align with the broader goal of mitigating bias in the peer review process. However, the main distinction between ours and previous works is that our method is parameter-free and operates without historical data. In contrast, previous studies typically need to fit model parameters or access to historical data.
>
> Regarding the "reducing bias by second-order information" category, our study tackles a problem distinct from those addressed in previous works. They primarily concentrate on the aggregation of forecasts. The settings and methodologies of most works cannot be extended to comparing paper quality in peer review contexts.
>
> The Surprisingly Popular (SP) method can be applied in our setting [3]. The SP method calculates the "prediction-normalized vote" for all signals $i$ as $\frac{w_i}{q_i}$ and asserts that the signal $\arg\max_i \frac{w_i}{q_i}$ is the best signal. Thus, when signals are binary ('reject': 0, 'accept': 1), the SP score of a paper is given by the "prediction-normalized vote" of 'accept' minus that of 'reject', i.e., $\frac{w_1}{q_1}-\frac{w_0}{q_0}$. However, the SP score is not invariant to noise. In some instances, even with infinite reviewers, the SP score fails to determine the relative quality of two papers.
>
> [3] Prelec, Dražen, H. Sebastian Seung, and John McCoy. "A solution to the single-question crowd wisdom problem." *Nature* 541.7638 (2017): 532-535.
>
> **An example that the SP score fails to work**
>
> * Signals are binary ('reject'/'accept').
> * The prior over the states is $w_1\sim \text{Uniform}(0,1)$ (paper’s quality is distributed uniformly).
> * Paper A has true state $\mathbf{w}_A=[1/3,2/3]$, i.e., paper A has true quality $2/3$.
> * Paper A receives no noise, i.e., the noise matrix of paper A is $\mathbf{M}_A=\left[\\begin{matrix}1 & 0\\\\0 & 1\\\\ \\end{matrix}\right]$.
> * Paper B has true state $\mathbf{w}_B=[1/4,3/4]$, i.e., paper B has true quality $3/4$, which is better than paper A.
> * Paper B receives positive noise. The noise matrix of paper B is $\mathbf{M}_B=\left[\\begin{matrix}1/3 & 2/3\\\\0 & 1\\\\ \\end{matrix}\right]$.
>
> For paper A, $\hat{\mathbf{w}}_A=[1/3,2/3]\, \hat{\mathbf{q}}_A=[1/2,1/2]$. The SP score of paper A is $\frac{2/3}{1/2}-\frac{1/3}{1/2}=\frac{2}{3}$.
>
> For paper B, $\hat{\mathbf{w}}_B=[1/12,11/12]\, \hat{\mathbf{q}}_B=[1/6,5/6]$. The SP score of paper B is $\frac{11/12}{5/6}-\frac{1/12}{1/6}=\frac{3}{5}$.
>
> As a result, paper B has a better true quality than paper A, but has a lower SP score, even if there are infinite reviewers.
>
>
> ### Q2: Purpose of the invariant surprisal vector
>
> The core idea is that a metric can facilitate quantitative comparisons across different papers only if it is invariant to systematically biased noise. Without invariance, comparisons for certain papers can easily be skewed by biased noise. For instance, with one noise pattern, paper A outscore paper B, while with a different biased noise pattern, paper B outscore paper A.
>
> Specifically, the invariant surprisal vector forms the main building block for the design of the surprisal-based score. This score effectively compares papers as if we had their true quality, especially as the number of reviewers goes to infinity. This property crucially relies on the invariance. We will clarify it in the final version.
>
>
> ### Q3: Could the authors clarify the aspects of Section 1 and 2
>
> **Clarification of Example 2**
>
> In Example 2, we want to express that "a higher noise level would result in a lower expected score for high-quality papers", rather than "a higher noise level would result in a lower expected score".
>
> In popular topics, the abundance of expert reviewers ensures a diminished noise in the review process. This ensures that the evaluations from reviewers align closely with the true paper quality. Conversely, in the topics where expert reviewers are scarce, the review process is prone to conservative noise. Consequently, there's a higher probability for reviewers' evaluations to deviate from the actual quality of the papers.
>
> **Clarification of which aspects are observable by agents and by the mechanism**
>
> In brief, each paper has a true state $\mathbf{w}$ representing its actual quality (nobody knows). Reviewer $i$ is aware of the prior distribution $\mathbf{Q}$ over $\mathbf{w}$ and receives a signal $x_i$ reflective of the paper's true state $\mathbf{w}$. Using both the prior $\mathbf{Q}$ and the signal $x_i$, reviewer $i$ forms a posterior distribution on $\mathbf{w}$ to anticipate the evaluations of fellow reviewers, represented as $\mathbf{q}_i$. The mechanism aggregates all signals and predictions $(\mathbf{q}_i, x_i)$ to assign the paper a score $S^*$. In the final version, we will relocate the symbol clarifications (Table 1, currently in the appendix on page 17) back to section 2 for ease of reference.

---

> > ### Comment · Reviewer_6t8A · 2023-08-14
> > **Response to Authors**
> >
> > Thank you for your thorough response and clarifications.
> > - The example showing the failure of the SP mechanism was helpful. Since this mechanism is applicable here, I think it would be good to add SP to the experimental comparison as an alternative baseline (or to explain why such a comparison wouldn't be useful).
> > - Regarding example 2: My understanding of the example is still that that bias in the noise is the issue ("conservative noise"): i.e., if the paper with non-expert reviewers has additional zero-mean noise in the ratings, this wouldn't change the expected fraction of accepts. In the paper, only the term "more noisy ratings" is used without mentioning the bias, which I think is more commonly interpreted as zero-mean noise. This is particularly confusing since the following example 3 specifically focuses on biased noise.
> > - I agree that moving the symbol table to the main paper would help improve the clarity.
> >
> > After reading the other reviews and the author response, I will improve my score from a 4 to a 5 for the time being. The contextualization of the work within the literature was helpful, and I hope that the proposed changes will improve the clarity. Despite this, my main concern about the presentation of the work still stands, and I still remain somewhat unsure about the significance of the results.

---

> > > ### Author Response · Authors · 2023-08-14
> > >
> > > ### Using SP for comparison
> > > Thank you for your feedback!
> > > The method proposed by Prelec et al. (2017) was designed for aggregating multiple reports into a single decision, rather than generating scores for comparing two alternatives. Specifically, when signals are binary, their approach outputs 1 if $\frac{w_1}{q_1}>\frac{w_0}{q_0}$, and 0 otherwise. While effective for combining evaluations into a single decision, this does not directly produce scores that can be used to compare the relative quality of two papers or alternatives.
> > >
> > > To enable such comparisons, we propose a new score based on the surprisingly popular idea: $\frac{w_1}{q_1}-\frac{w_0}{q_0}$: this measures the amount of surprisal for 'accept', subtracting the amount of surprisal for 'reject'. We call it the SP-inspired score.
> > > We run the same numerical experiments as Figure 4 and 5. Since there is no place to attach figures, we describe the simulation results in words.
> > > * SP-inspired score and the surprisal based score have similar performance in the settings considered by Figures 4 and 5 when the number of reviewers is small like $n=3,5$, and the surprisal based score outperforms the SP-inspired score as the number of reviewers increase. Both of them are better than the simple average.
> > >
> > > Conceptually, the example we provided before demonstrates that the SP-inspired score does not calibrate based on the amount of noise. Without calibrating for noise amount, the SP-inspired score will introduce substantial bias against papers with high-quality but more noisy evaluations. Because of the lack of noise calibration, the error rate of the SP-inspired score does not converge to zero as the number of reviewers increases. This makes it challenging to provide theoretical guarantees on the SP-inspired score's performance across all cases. In contrast, the surprisal-based score does calibrate, allowing its error rate to decrease with more reviewers, backed by theoretical guarantees.
> > >
> > > However, without calibration, while suffering from bias, the SP-inspired score has a lower variance. When reviewer numbers are small, the SP-inspired score achieves similar average performance to the surprisal-based score.
> > >
> > > In the final version, we will include the SP-inspired score and expanded comparison results to further illustrate the strengths and limitations of both methods.
> > >
> > > ### Example 2
> > >
> > > Thanks for pointing it out. We will clarify this in Example 2. The non-experts reviewers have challenges to evaluate technical quality and novelty, but have access to superficial cheap signals. To reflect these limitations, in our setting, the non-expert reviews are modeled not as zero-mean noise, but as biased towards the prior rating for papers.

---

### Official Review · Reviewer_UucL · 2023-07-09

**Soundness:** 2 fair
**Presentation:** 2 fair
**Contribution:** 2 fair
**Rating:** 4
**Confidence:** 3

**Summary:**

This paper aims to detect and correct bias in Peer Review. They propose a one-shot noise calibration process without any prior information. Experiments are conducted on the binary case to show the effectiveness of the proposed method.

**Strengths:**

1. The studied problem is important.
2. Theoretical guarantee is provided for the proposed calibrated score.
3. The organization of this paper is clear.


**Weaknesses:**

1. The rationality of the proposed surprisal scores needs further support. For example, in Figure 1, it seems that the left paper with lower negative reviewers’ prediction is the same as the right paper with higher negative reviewers’ prediction, while the left paper receives 1 accept and 2 reject while the right paper receives 2 accept and 1 reject. This result is confusing since the reviewers may give low P_0,1 simply due to the poor quality of the left paper.
2. The experiments are weak in demonstrating the generality of the proposed method. For example, in Sec 6, the authors only conduct experiments on the binary case, lacking the more general settings with more types of possible signals.
3. The theoretical guarantees lack a clear explanation. For example, in theorem 2, why there is 1/2 in the error probability (Pr[S(A) > S(B)|w1^A,w2^B]+1/2 Pr[S(A)=S(B)|w1^A，w1^B])? The guarantee of such an error probability may still remain a gap in the performance guarantee of the proposed method.
4. Some assumption or claims lacks further support and needs to offer more clarification or explanation. For example, in Sec 2.2, why “only consider the noise where M is invertible?” What will be the situation when M is non-invertible? Why can construct the vector q and the joint distribution matrix U from the prediction matrix P as stated in claim 1?  Is it universally applicable to assume that 'each individual receives the clean signal with probability 1-\lambda’?


**Questions:**

1. In theorem 2, why there is 1/2 in the error probability (Pr[S(A) > S(B)|w1^A,w2^B]+1/2 Pr[S(A)=S(B)|w1^A，w1^B])? The guarantee of such an error probability may still remain a gap in the performance guarantee of the proposed method.
2. In Sec 2.2, why “only consider the noise where M is invertible”? What will be the situation when M is non-invertible?
3. Why can construct the vector q and the joint distribution matrix U from the prediction matrix P as stated in claim 1?
4. Is it universally applicable to assume that 'each individual receives the clean signal with probability 1-\lambda’?


**Limitations:**

The authors have discussed the limitations of this paper in Sec 7.

---

> ### Author Rebuttal · Authors · 2023-08-10
>
> ## Reviewer UucL
>
> Thank you for your insightful comments and suggestions.
>
> ### W1: Rationality of the proposed surprisal scores
>
> Our understanding of your question suggests that you are seeking a more intuitive explanation of Figure 1, especially concerning why our method yield identical scores in the following two scenarios:
>
> * Scenario A: 1 accept and 2 rejects, where the 'accept' reviewer predicts $P_{1,1}=0.55$ and the 'reject' reviewers predict $P_{0,1}\approx 0.07$.
> * Scenario B: 2 accepts and 1 reject, where the 'accept' reviewers predict $P_{1,1}=0.55$ and the 'reject' reviewer predicts $P_{0,1}\approx 0.4$.
>
> In Scenario A, the 'reject' reviewers anticipated that very few would vote for 'accept'. However, actually 1/3 of the reviewers vote for 'accept', making 'accept' a surprisingly popular signal. This suggests that the reviewers might have been swayed by noise skewing towards 'reject'. In contrast, Scenario 2 witnessed 'accept' votes aligning closely with the predictions, suggesting minimal interference from noise. According to our model, when a voting outcome exceeds the crowd's predictions, it is more likely to be accurate. This reasoning accounts for the identical scores assigned to both scenarios.
>
>
> ### Q1(1): In theorem 2, why there is 1/2 in the error probability (Pr[S(A) > S(B)|w1^A,w2^B]+1/2 Pr[S(A)=S(B)|w1^A，w1^B])?
>
> The occurrence of 1/2 arises when we use random selection to break ties when comparing two papers that have the same scores. For instance, a tie might arise when all reviewers vote for "accept" to both papers (scores being set to $+\infty$). In these situations, there is no additional information to compare the quality of two papers, which induces 1/2 in error probability.
>
> In real-world scenarios, we typically only compare papers that have received at least one "accept" and one "reject" evaluation. This is because papers with unanimous 'accept' evaluations are generally accepted, while those with all 'reject' evaluations are rejected. Given this practice, the likelihood of two papers garnering the same score is low.
>
>
> ### Q1(2): The guarantee of such an error probability remains a gap in the performance guarantee of the proposed method.
>
> The main reason of the gap is that our theoretical guarantees encompass all conceivable scenarios, but simulation experiments evaluate only a finite set of situations. It is crucial to underscore the importance of theoretical guarantees, as they validate the effectiveness of our mechanism. By 'effectiveness', we mean that given a sufficient number of reviewers, our method can attain arbitrarily low error rates; even with a limited number of reviewers, the error rate remains constrained and predictable.
>
>
> ### Q2: Why only consider the noise where M is invertible
>
> The assumption of "invertible correlation" is prevalent in peer prediction literature [1-2]. In our paper, the term "invertible noise" (i.e., noise matrix $M$ is invertible) signifies an "invertible correlation" between each reviewer's received signal and their clean signal. In the binary case, if $M$ is non-invertible, the reviewer's received signal and clean signal become independent, rendering the received signal uninformative. For general cases, the non-invertibility of $M$ implies that there exists two papers with distinct true qualities but receive the same evaluation distribution from reviewers. Such facts prevent the existence of an effective aggregation method when noise is non-invertible.
>
> [1] Kong, Yuqing. "Dominantly truthful multi-task peer prediction with a constant number of tasks." Proceedings of the fourteenth annual acm-siam symposium on discrete algorithms. Society for Industrial and Applied Mathematics, 2020.
>
> [2] Schoenebeck, Grant, and Fang-Yi Yu. "Learning and Strongly Truthful Multi-Task Peer Prediction: A Variational Approach." 12th Innovations in Theoretical Computer Science Conference (ITCS 2021). Schloss Dagstuhl-Leibniz-Zentrum für Informatik, 2021.
>
>
> ### Q3: Why can construct the vector q and the joint distribution matrix U from the prediction matrix P as stated in claim 1
>
> Claim 1 in this paper is initially introduced by Prelec et al. in the context of the Surprisingly Popular (SP) method [3]. Here we briefly delve into the theorem's intuition. Applying Bayes' rule, we deduce $q_t=q_s\frac{P_{s,t}}{P_{t,s}}$. Given that $\sum_t q_t = 1$, we can solve for the prior probability of $q_s$: $q_s = (\sum_{t}\frac{P_{s,t}}{P_{t,s}})^{-1}$. Regarding the joint distribution $U$, by definition, we have $U_{s,t}=q_s P_{s,t}$.
>
> [3] Prelec, D., Seung, H. & McCoy, J. A solution to the single-question crowd wisdom problem. Nature 541, 532–535 (2017).
>
>
> ### Q4: Is it universally applicable to assume that each individual receives the clean signal with probability 1-lambda?
>
> In the paper we define $\mathcal{M}^*$ as the noise family where an individual receives the clean signal with a probability of $1-\lambda$ and a biased value (which is independent of quality) with a probability of $\lambda$. In Claim 2, we establish that $\mathcal{M}^*$ encompasses all positively correlated ($M_{1,1} > M_{0,1}$) and invertible noises in the context of binary signals. This emphasizes that $\mathcal{M}^*$ serves as a universal noise model for binary signals. However, in general settings, $\mathcal{M}^*$ is not comprehensive enough to describe all potential noise. In Appendix B, we introduce a modified metric that remains invariant across all possible invertible noises. We also demonstrate its capability in identifying the true state $\mathbf{w}$ under specific circumstances.

---

### Official Review · Reviewer_SV22 · 2023-07-14

**Soundness:** 3 good
**Presentation:** 3 good
**Contribution:** 2 fair
**Rating:** 5
**Confidence:** 3

**Summary:**

The paper tries to solve famous problem of removing bias and noise during peer-review process. The authors address the problem in one-shot setting (without historical data) and propose a novel approach that allows to remove influence of bias and noise (under some assumptions on bias and noise) on the ranking of assessed elements (reviewed papers). The approach uses scoring adjustment (used for ranking) and requires additional signal (action) from each reviewer.
The authors provide theoretical guarantees and small synthetic experiments.

**Strengths:**

-	Overall good organization of the paper storyline (very good Intro!)
-	More or less clear statements and easy to follow
-	Synthetic experiments

**Weaknesses:**

-	Some drawbacks in clarity related to assumptions and limitations of the setup
-	Seem limited practical effect
-	The contribution does not look very enough for NeurIPS
-	Lack of comparison with alternative ways to improve scoring (e.g., non-1-shot ones, etc)

**Questions:**

1. It would be nice to understand whether the assumptions are realistic (for instance, Lines 59-60 “the correct answer is positively correlated to each agent’s signal” ; Lines 82 83 “assume that the noise is positively correlated .. ”; and the other). Right now the assumptions look theoretical with no clear understanding, e,g., how frequent they are in practice.

2.  The specific assumption about particular cases of noises (Lines 203-205) looks very important while being deferred to Sec3 as Definition 1 (instead of being kept in Section 2 where environment of the problem is described). It makes feeling of overclaiming in Intro, where our Results (Lines 85-91) are stated without assumption of considering invertible noise (while other assumptions and limitations are addressed before Line 85). It would be nice to have some presentation improvements to address this gap.

3. The authors claim that it is nice to have the result about best ranking when the number of reviewers goes to infinity. However, I believe it is not so important result to initially stated problem, where the limitations on the number of reviewers is crucial part of the problem (in most peer review practice we deal with 3-5 reviewers, e.g., see the case of NeurIPS). So, OK to have such theorem, but this result seems not so important and relevant to the problem taken for the research. It would be nice, if the authors focus at least the presentation more on how their results help in real practical cases like n = 3-5. (Yes, I see that the authors are aware of such numbers by providing experiments with such numbers. But (a) it might be just because of simplification of experimentation; and (b) the main contribution lies in the theoretical part of the work).

4. While reading from the first page till the end, I had observed changing perception (for sure, mine, subjective) of the studied setting. From the beginning of the paper till Page 6 (Lines 189-190), I thought that the setup assumes that all reviewers assess all papers, while, in practice, each agent review few numbers of papers. Sections 2.1 2.2 have made me sure that the setup is so strange. Only, Lines 189+ reverted my perception. So, I would strongly recommend to somehow help such readers as I figure out this earlier than Page 6.

5.The statement in Lines 168-169 «we focus on homogeneous noise setting where the noise is the same for all reviewers and will discuss the heterogeneous setting in Section 7" makes feeling of covering “heterogeneous setting” by this work. After reaching Section 7, I have found that Section 7 is Conclusion section and heterogeneous setting are discussed as some future work... Missed expectation

========

AFTER REBUTTAL
I thank the authors for answering the questions. I hope that the answers will be reflected and fully addressed in the new revision of the work.

**Limitations:**

Some presentation improvements may address limitations better
See points 1, 2 and 3 in the field “Questions”.

---

> ### Author Rebuttal · Authors · 2023-08-10
>
> ## Reviewer SV22
>
> Thank you for your insightful comments and suggestions.
>
> ### Q1: Whether the assumptions are realistic
>
> **Positively correlated**
>
> In the assumptions mentioned in lines 59-60, "positively correlated" indicates that the correct answer has a positive correlation with the signal, as discussed in [1]. For example, if the signal is "yes", the posterior belief that the correct answer is "yes" would be greater than the prior belief. In our model (lines 82-83), the positive correlation, denoted as $M_{1,1}>M_{0,1}$, indicates that the reviewer's clean signal correlates positively with her received noisy signal. This assumption is realistic, as a negative correlation would imply that if the reviewer receives 'accept', her posterior belief regarding her clean signal (which she can ascertain with full effort) being "accept" would be lower than the prior belief, which is illogical. We will clarify the distinction of "positively correlated" in lines 59-60 and 82-83 in the final version.
>
> **Non-degenerate noise**
>
> The assumption of "invertible correlation" is prevalent in peer prediction literature [2-3]. In our paper, the assumption of "non-degenerate noise" in lines 82-83 (and "invertible noise" in line 206) means a "invertible correlation" between each reviewer's received signal and their clean signal. In the binary case, if $M$ is non-invertible, the reviewer's received signal and clean signal become independent, rendering the received signal uninformative. In general cases, the non-invertibility of $M$ implies that there exists two papers with distinct true qualities but receive identical evaluation distribution from reviewers. Such facts prevent the existence of an effective aggregation method when noise is non-invertible.
>
> In the final version, we will standardize the terms "non-degenerate noise" and "invertible noise" to consistently use "non-degenerate noise".
>
> [1] Prelec, D., Seung, H. & McCoy, J. A solution to the single-question crowd wisdom problem. *Nature* **541**, 532–535 (2017).
>
> [2] Kong, Yuqing. "Dominantly truthful multi-task peer prediction with a constant number of tasks." Proceedings of the fourteenth annual acm-siam symposium on discrete algorithms. SODA 2020.
>
> [3] Schoenebeck, Grant, and Fang-Yi Yu. "Learning and Strongly Truthful Multi-Task Peer Prediction: A Variational Approach." ITCS 2021.
>
>
> ### Q2: Results (Lines 85-91) are stated without assumption of considering invertible noise
>
> The results (lines 82-91) are stated under the assumption that the noise is non-degenerate. In our paper, the term "non-degenerate" is synonymous with "invertible". In the final version, we will standardize the terms "non-degenerate noise" and "invertible noise" to consistently use "non-degenerate noise", and amend the phrase "Within the model, .." in line 85 to "Within the model and assumptions, ..". Thank you for pointing this out.
>
>
> ### Q3: How the results help in real practical cases like 3-5 reviewers
>
> Theorem 2 offers results for finite values of $n_A$ and $n_B$. Here, $n_A$ represents the number of reviewers for paper A, while $n_B$ represents the number of reviewers for paper B. Consequently, Theorem 2 establishes an upper bound on the error probability for realistic practical cases, such as when $n=3,5$.
>
> For a more practical perspective, consider this: usually, we only wish to compare papers that have received at least one "accept" and one "reject" evaluation. This is because papers with unanimous 'accept' evaluations are typically accepted, while those with all 'reject' evaluations are rejected. In such scenarios, the theoretical upper bound can be simplified to: $\exp\left(-\frac{2(w_1^B-w_1^A)^2}{\frac{1}{n_A(1-\lambda_A)^2}+\frac{1}{n_B(1-\lambda_B)^2}}\right)$. Here, $w_1^B-w_1^A$ signifies the average disparity in true quality, and $\frac{1}{n_A(1-\lambda_A)^2}+\frac{1}{n_B(1-\lambda_B)^2}$ relates to the number of reviewers and the maximum bias.
>
> This bound underscores that our methodology's error rate diminishes as the gap in true quality grows more pronounced. For example, when there are 5 reviewers with a maximum bias of 0.3, the error bound approximates to $\exp\left(-2.5|w_1^B-w_1^A|\right)$. The bound is reasonable, especially given that the bias can be arbitrary. To further elucidate, in Section 6, we incorporate simulation experiments to compare our approach with the baseline.
>
>
> ### Q4: Observed Changing perception of the studied setting
>
> Thank you for the valuable feedback. We will specify within Example 1 that reviewers have the flexibility to assess any number of papers, and that different papers may be subject to varying biases.
>
>
> ### Q5: Discussion of heterogeneous reviewer noise
>
> In one-shot settings, tackling heterogeneous noise is challenging, primarily because the mechanism has no access to historical data from reviewers. Moreover, each reviewer typically assesses only a few papers during the review cycle. While existing research on peer grading employs Gibbs sampling to gauge the bias of each evaluator, this method necessitates a significant number of evaluations from each evaluator [4-5]. Practically, certain techniques can mitigate noise at the reviewer level. For instance, using benchmark papers can help calibrate reviewers' evaluations. In our final version of the paper, we will reintroduce the relevant discussion that was previously condensed due to page limit.
>
> [4] Piech, Chris, et al. "Tuned models of peer assessment in MOOCs.".
>
> [5] Mi, Fei, and Dit-Yan Yeung. "Probabilistic graphical models for boosting cardinal and ordinal peer grading in MOOCs." *Proceedings of the AAAI Conference on Artificial Intelligence*. Vol. 29. No. 1. 2015.

---

### Author Rebuttal · Authors · 2023-08-10

We thank the reviewers for their insightful suggestions and comments on our manuscript. We are pleased that the reviewers recognize the novelty (Reviewer SV22, tv7v, VrqZ), clarity in writing and organization (Reviewer SV22, UucL, tv7v), as well as the sound theoretical guarantees and synthetic experiments (Reviewer SV22, UucL, 6t8A). We will address the inquiries and comments from each reviewer separately.

---

### Decision · Program_Chairs · 2023-09-21

**Decision:**

Accept (poster)

**Comment:**

This paper proposes a method to mitigate bias and noise during peer-review process.
The authors present theoretical guarantees in a simplified setting, accompanied by synthetic experiments.

Reviewers agree that the paper is very well-motivated. The topic is important, and has been gaining traction recently, especially in light of the state of reviewing in the machine learning community. The primary concern reviewers have about this paper lie in the strength of the assumptions. However, this topic is relatively young, and so it benefits from this type of simplified analysis, to better understand the problem parameters in tractable settings. I thus recommend acceptance. I believe this paper will be an insightful step in the quest for better peer review mechanisms.

I encourage the authors to take into account the feedback from reviewers in the camera-ready. Specifically, several reviewers noted that the later sections are overly general, with unclear payoff for the reader – and I tend to agree.